# CNKSR1 serves as a scaffold to activate an EGFR phosphatase via exclusive interaction with RhoB-GTP

Kanako Nishiyama[1,2,*], Masashi Maekawa[2,3,*], Tomoya Nakagita[4], Jun Nakayama[5], Takeshi Kiyoi[6], Mami Chosei[3], Akari Murakami[1], Yoshiaki Kamei[1], Hiroyuki Takeda[4], Yasutsugu Takada[1], Shigeki Higashiyama[2,3,7]

Epidermal growth factor receptor (EGFR) and human EGFR 2 (HER2) phosphorylation drives HER2-positive breast cancer cell proliferation. Enforced activation of phosphatases for those receptors could be a therapeutic option for HER2-positive breast cancers. Here, we report that degradation of an endosomal small GTPase, RhoB, by the ubiquitin ligase complex cullin-3 (CUL3)/KCTD10 is essential for both EGFR and HER2 phosphorylation in HER2-positive breast cancer cells. Using human protein arrays produced in a wheat cell-free protein synthesis system, RhoB-GTP, and protein tyrosine phosphatase receptor type H (PTPRH) were identified as interacting proteins of connector enhancer of kinase suppressor of Ras1 (CNKSR1). Mechanistically, constitutive degradation of RhoB, which is mediated by the CUL3/KCTD10 E3 complex, enabled CNKSR1 to interact with PTPRH at the plasma membrane resulting in inactivation of EGFR phosphatase activity. Depletion of CUL3 or KCTD10 led to the accumulation of RhoB-GTP at the plasma membrane followed by its interaction with CNKSR1, which released activated PTPRH from CNKSR1. This study suggests a mechanism of PTPRH activation through the exclusive binding of RhoB-GTP to CNKSR1.

## Introduction

Inactivation of receptor tyrosine kinases is mainly achieved through their dephosphorylation by protein tyrosine phosphatases. Thus, protein tyrosine phosphatases generally act as tumor suppressors by terminating signal transduction from phosphorylated oncogenic receptor tyrosine kinases (Bollu et al, 2017). Exceptionally, several protein tyrosine phosphatases such as PTPN11 (alias SHP2) and PTP4A3 function as oncogenes (Bunda et al, 2015; den Hollander et al, 2016). Another protein tyrosine phosphatase, PTPN1, acts as a tumor suppressor in lymphomagenesis or an oncoprotein in breast

cancers and non–small cell lung cancers (Dubé et al, 2005; Julien et al, 2007).

Human epidermal growth factor receptor (EGFR) 2 (HER2) is a member of EGFR tyrosine kinase family. Unlike other EGFR family members (EGFR, HER3, and HER4), HER2 recognizes no known ligands and forms heterodimers with each other member, leading to the transduction of various cellular signals (Slamon et al, 1989; Yarden & Sliwkowski, 2001; Moasser, 2007). Especially, the activated EGFR transactivates HER2, and the phosphorylation of both EGFR and HER2 strongly drives cell proliferation (Stern & Kamps, 1988; Kokai et al, 1989). Pathologically, HER2-positive breast cancers account for 15–20% of human breast cancers and are more aggressive than other types of breast cancers owing to their high proliferative and metastatic capacities (Dai et al, 2016). Although HER2-targeted therapeutics, such as the administration of an inhibitory humanized monoclonal antibody against HER2 (e.g., trastuzumab), are effective for HER2-positive breast cancer patients, the development of trastuzumab resistance with its long-term administration often poses a major challenge (Esteva et al, 2002; Nahta et al, 2006; Dokmanovic et al, 2011). Because the activation of protein tyrosine phosphatases could principally reduce the phosphorylation of receptor tyrosine kinases leading to the suppression of cell growth, the forced activation of protein tyrosine phosphatases targeting EGFR and/or HER2 could be new strategy for HER2-positive breast cancer therapies.

Cullin-3 (CUL3) is a scaffold protein for cullin/RING-type E3 ubiquitin ligase complexes. In such complexes, a substrate recognition receptor, specifically a BTB domain-containing protein (BTBP), recruits substrates leading to their ubiquitination (Petroski & Deshaies, 2005). One BTBP, KCTD10, recognizes various substrates (e.g., RhoB, CEP97, EIF3D, TRIF, and EPS8) (Kovačević et al, 2018; Nagai et al, 2018; Maekawa et al, 2019; Murakami et al, 2019; Wu et al, 2019; Maekawa & Higashiyama, 2020; Rodríguez-Pérez et al, 2021). We previously reported that the constitutive degradation of an endosomal small GTPase RhoB by the CUL3/KCTD10 E3 complex functions

[1]Department of Hepato-Biliary-Pancreatic Surgery and Breast Surgery, Ehime University Graduate School of Medicine, Toon, Japan [2]Department of Biochemistry and Molecular Genetics, Ehime University Graduate School of Medicine, Toon, Japan [3]Division of Cell Growth and Tumor Regulation, Proteo-Science Center, Ehime University, Toon, Japan [4]Division of Proteo-Drug-Discovery Sciences, Proteo-Science Center, Ehime University, Matsuyama, Japan [5]Division of Cellular Signaling, National Cancer Center Research Institute, Chuo-ku, Japan [6]Division of Analytical Bio-medicine, Advanced Research Support Center, Ehime University, Toon, Japan [7]Department of Molecular and Cellular Biology, Osaka International Cancer Institute, Chuo-ku, Osaka, Japan

Correspondence: masashim@m.ehime-u.ac.jp; shigeki@m.ehime-u.ac.jp
*Kanako Nishiyama and Masashi Maekawa contributed equally to this work

in EGF-induced Rac1 activation specifically in HER2-positive breast cancer cell lines, among other breast cancer subtypes (Murakami et al, 2019). However, the roles of CUL3/KCTD10 E3 complex in EGFR/HER2 signaling pathways remain unclear.

In this study, we found that depletion of CUL3 or KCTD10 reduced the phosphorylation of EGFR and HER2, as well as cell proliferation of HER2-positive breast cancer cells. Making use of a human protein array (4,212 proteins) and a human phosphatase array (171 phosphatases) produced in a wheat cell-free protein synthesis system, we found that connector enhancer of kinase suppressor of Ras1 (CNKSR1) directly interacted with RhoB-GTP and protein tyrosine phosphatase receptor type H (PTPRH). In cells depleted of CUL3 or KCTD10, accumulated RhoB-GTP released an EGFR phosphatase, PTPRH, from a scaffold protein, CNKSR1, via its exclusive binding to the PH domain of CNKSR1 at the plasma membrane, resulting in PTPRH activation. Our results suggest a unique model of phosphatase activation by protein–protein interactions at the plasma membrane.

## Results

### The CUL3/KCTD10 E3 complex is essential for EGFR and HER2 phosphorylation through the degradation of RhoB in HER2-positive breast cancer cells

To examine the roles of the CUL3/KCTD10 complex in the activation of EGFR and HER2, we first examined the phosphorylation of those receptors in HER2-positive breast cancer cells. The phosphorylation of $Tyr^{1068}$ in EGFR and $Tyr^{1221/1222}$ in HER2 was detected in a HER2-positive breast cancer cell line, SKBR-3 cells, without the addition of any ligands of the HER family (Fig 1A). These data suggested that EGFR is activated by para- and/or autocrine ligands and that HER2 is transactivated by the activated EGFR in SKBR-3 cells (Stern & Kamps, 1988; Kokai et al, 1989). Knockdown of CUL3 or KCTD10 reduced the phosphorylation of both EGFR (ratio of pEGFR/EGFR) and HER2 (ratio of pHER2/HER2) (Fig 1A). The expression of siRNA-resistant CUL3 or KCTD10 in CUL3- or KCTD10-depleted SKBR-3 cells, respectively, restored the decreased phosphorylation of both EGFR and HER2, excluding any off-target effects of siRNA (Fig 1A). The CUL3/KCTD10 E3 complex recruits its substrate RhoB, an endosomal small GTPase, leading to polyubiquitination of RhoB, followed by its lysosomal degradation (Kovačević et al, 2018). As reported previously (Murakami et al, 2019), the knockdown of CUL3 or KCTD10 caused the accumulation of RhoB protein in SKBR-3 cells (Fig 1A). The knockdown of RhoB in CUL3- or KCTD10-depleted SKBR-3 cells partially restored the decreased phosphorylation of both EGFR and HER2 (Fig 1B). In contrast, RhoB knockdown increased the phosphorylation of both EGFR and HER2 in SKBR-3 cells (Fig 1C). GTP-bound small GTPases function as active forms through interactions with various effector proteins. We found that RhoB-GTP, as well as total RhoB protein, was accumulated in CUL3- or KCTD10-knockdown cells (Fig 1D). Protein expression of other Rho GTPases, such as RhoA and RhoC, was not affected by CUL3 or KCTD10 knockdown, and GTP forms of RhoA and RhoC were detected in neither control, CUL3-, nor KCTD10-depleted SKBR-3 cells (Fig 1D).

Treatment of CUL3- or KCTD10-depleted SKBR-3 cells with C3 transferase, which inhibits the conversion of Rho-GDP to Rho-GTP, restored the decreased phosphorylation of both EGFR and HER2 (Fig 1E). The knockdown of CUL3 or KCTD10 significantly reduced the proliferation of SKBR-3 cells in the absence of exogenous growth factors (Fig 1F). Treating SKBR-3 cells with MLN4924, a neddylation inhibitor that inhibits CUL3-mediated ubiquitination, caused the accumulation of RhoB protein and reduced the phosphorylation of both EGFR and HER2 (Fig 1G). Taken together, the constitutive degradation of RhoB by CUL3/KCTD10 is essential for the phosphorylation of both EGFR and HER2, as well as the proliferation of SKBR-3 cells, and the accumulation of RhoB-GTP inhibits the phosphorylation of both EGFR and HER2 (Fig 1H).

### CNKSR1, an interacting protein of RhoB-GTP, positively regulates proliferative signaling in HER2-positive breast cancer cells

We next sought to identify proteins that directly interact with RhoB-GTP and regulate EGFR and HER2 phosphorylation in SKBR-3 cells. For this aim, we produced biotinylated recombinant proteins of wild-type RhoB, a constitutively active RhoB mutant (Q63L), and a dominant negative RhoB mutant (T19N) using a wheat cell-free protein synthesis system (Sawasaki et al, 2002, 2008) (Fig 2A). The biotinylated RhoB proteins (bait proteins) were then incubated with a FLAG-GST-tagged 4,212 human protein array aliquoted in 11 plates of 384-well plate, respectively, followed by AlphaScreen protein–protein interaction screen (Fig 2B). We detected enhanced luminescence signals between biotinylated RhoB (Q63L), which mimics RhoB-GTP, and RTKN (alias Rhotekin), which has been shown to interact with RhoB-GTP (Reid et al, 1996), indicating the reliability of this screen (Fig 2C and Table S1). Among the RhoB-GTP-interacting proteins in the screen, we focused on CNKSR1, which specifically interacted with RhoB (Q63L) mutant but not wild-type RhoB and RhoB (T19N) (Fig 2C). CNKSR1 is a scaffold protein that possesses five domains in its amino acid sequence (Sundaram & Han, 1995; Therrien et al, 1995; Jaffe et al, 2004) (Fig 2D) and positively regulates cell proliferative signals (Fritz & Radziwill, 2011). To confirm the direct interaction between RhoB (Q63L) and CNKSR1 by AlphaScreen, we produced various mutants and deletion forms of both proteins using a wheat cell-free protein synthesis system (Fig 2E). FLAG-tagged RhoB (Q63L), but not wild-type RhoB and RhoB (T19N), directly interacted with biotinylated CNKSR1, excluding the effects of tags (Fig 2F). Enhanced luminescence signals between RhoB (Q63L) and CNKSR1 were not detected by deletion of the PH domain of CNKSR1 (Fig 2G), and the PH domain of CNKSR1 itself directly interacted with RhoB (Q63L) (Fig 2H). Among the PH domain mutants, we detected the remarkably increased luminescence signals between RhoB (Q63L) and the K414E/P416E/R423A/R425L/R426A/K478N (EEALAN) mutant which lacks affinity for a phosphoinositide, phosphatidylinositol 4,5 bisphosphate (PI[4,5]$P_2$) (Indarte et al, 2019) (Fig 2H). It is likely that the introduction of six point-mutations in the PH domain causes its conformational changes, by which RhoB-GTP may easily access to its binding pocket in the EEALAN mutant. The addition of PI(4,5)$P_2$ did not inhibit the interaction between RhoB (Q63L) and CNKSR1 in vitro (Fig 2I). These data suggest that PI(4,5)$P_2$ does not interfere the interaction between RhoB-GTP and CNKSR1. Other PH domain mutants, K414R, which is not acetylated (Fischer

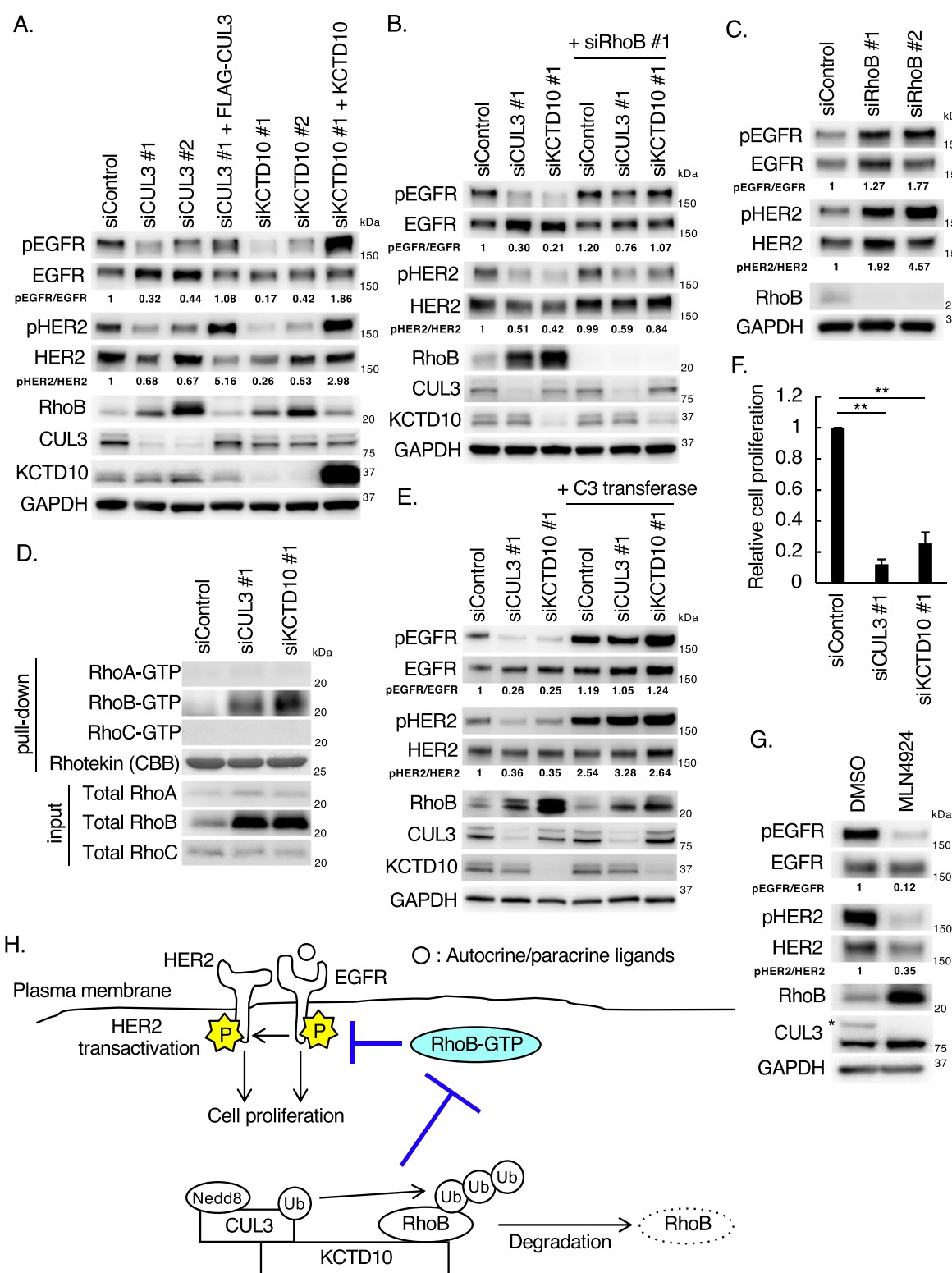

**Figure 1.   Knockdown of CUL3 or KCTD10 reduced the phosphorylation of epidermal growth factor receptor (EGFR) and HER2 through accumulation of RhoB-GTP in SKBR-3 cells.**
**(A)** Western blots of SKBR-3 lysates, 72 h post-transfection with the indicated siRNAs. Rescue experiments of CUL3 and KCTD10 knockdown by infection with siRNA-resistant-FLAG-CUL3-, or siRNA-resistant non-tagged KCTD10-carrying lentivirus. pEGFR, phosphorylated EGFR (Tyr[1068]); pHER2, phosphorylated HER2 (Tyr[1221/1222]). The ratio of band intensities for pEGFR/EGFR or pHER2/HER2, normalized to control, was shown. **(B, C)** Western blots of SKBR-3 lysates, 72 h post-transfection with the indicated siRNAs. pEGFR, phosphorylated EGFR (Tyr[1068]); pHER2, phosphorylated HER2 (Tyr[1221/1222]). The ratio of band intensities for pEGFR/EGFR or pHER2/HER2, normalized to

et al, 2017), and K414Q, which mimics the acetylated PH domain (Fischer et al, 2017), interacted with RhoB (Q63L), as was seen with the wild-type PH domain (Fig 2H). These data suggest that RhoB-GTP does not interfere association of acetylated CNKSR1 to the plasma membrane. In contrast, a W493A mutant of the PH domain (Jaffe et al, 2004) was completely devoid of luminescence signals with RhoB (Q63L) (Fig 2H). The introduction of the W493A mutation into full-length of CNKSR1 did not result in enhanced luminescence signals with RhoB (Q63L) (Fig 2J). Taken together, the Trp$^{493}$ residue in the PH domain of CNKSR1 is critical for the direct interaction between CNKSR1 and RhoB-GTP.

We then validated the interaction between RhoB-GTP and CNKSR1 in SKBR-3 cells. In control SKBR-3 cells in which RhoB-GTP is undetectable (Fig 1D), CNKSR1 localized to cytosol and membrane ruffles (Fig 3A; arrowheads). In contrast, in CUL3- or KCTD10-depleted SKBR-3 cells in which RhoB-GTP is accumulated (Fig 1D), RhoB and CNKSR1 were colocalized at the plasma membrane (Fig 3A). Expression of the Myc-tagged PH domain of CNKSR1 resulted in relocation of RhoB to the plasma membrane, and colocalization of the PH domain of CNKSR1 and RhoB at the plasma membrane was observed in control, CUL3-knockdown, or KCTD10-knockdown SKBR-3 cells (Fig 3B). Both the PH domain-deleted (ΔPH) mutant and W493A mutant of CNKSR1 localized to the cytosol, and colocalization with RhoB at the plasma membrane was diminished (Fig 3C and D). We also biochemically confirmed that the purified RhoB (Q63L) recombinant proteins interacted with wild-type CNKSR1 but not with ΔPH and W493A mutants of CNKSR1 derived from SKBR-3 cells (Figs 3E and F and S1). These data suggested that CNKSR1 interacts with RhoB-GTP at the plasma membrane in CUL3- or KCTD10-depleted cells in a PH domain-dependent manner.

To examine if CNKSR1 controls EGFR and HER2 phosphorylation, we next silenced CNKSR1 gene in SKBR-3 cells. CNKSR1 knockdown reduced the phosphorylation of EGFR (Fig 3G). Although the pHER2/HER2 ratio was slightly reduced by the knockdown of CNKSR1, the total amount of HER2 protein was decreased in CNKSR1-depleted SKBR-3 cells (Fig 3G). The proliferation of SKBR-3 cells was significantly inhibited by CNKSR1 knockdown (Fig 3H). We confirmed that the phosphorylation of Ser$^{473}$ in Akt and Thr$^{202}$/Tyr$^{204}$ in ERK1/2, downstream of EGFR/HER2 signaling, was decreased by CUL3, KCTD10, or CNKSR1 knockdown and increased by RhoB knockdown (Fig 3I). The phosphorylation of Tyr$^{1222}$ in HER3, another HER family protein, was also decreased by CUL3, KCTD10, or CNKSR1 knockdown and increased by RhoB knockdown (Fig 3J). The tyrosine kinase activities of HER3 are catalytically impaired (Guy et al, 1994; Jura et al, 2009). Thus, these data suggested that HER3 is transactivated by the activated EGFR or/and HER2 in control SKBR-3 cells, and

phosphorylation of HER3 is also controlled by the CUL3/KCTD10/RhoB/CNKSR1 axis as well as that of EGFR and HER2. The protein expression of HER4 was not detectable in SKBR-3 cells (Fig 3J). Collectively, CNKSR1 positively regulates proliferative signaling, including the phosphorylation of EGFR, in SKBR-3 cells.

### Interaction of CNKSR1 with RhoB-GTP is independent of PI(4,5)P$_2$ in HER2-positive breast cancer cells

The PH domain of CNKSR1 interacts with PI(4,5)P$_2$, a phosphoinositide which localizes at the cytosolic leaflets of the plasma membrane (Maekawa & Fairn, 2014; Indarte et al, 2019). The in silico screen of a compound library identified a CNKSR1 inhibitor, PHT-7.3, which binds to the PH domain and interferes the interaction between the PH domain and PI(4,5)P$_2$ (Indarte et al, 2019). The plasmalemmal localization of CNKSR1 as well as colocalization with K-Ras mutant at the plasma membrane was lost by treatment of K-Ras mutated cancer cells with PHT-7.3 (Indarte et al, 2019). We thus examined the effects of PI(4,5)P$_2$ on the interaction of CNKSR1 and RhoB-GTP at the plasma membrane of SKBR-3 cells. We found that GFP-PLC-PH, a biosensor of PI(4,5)P$_2$, localized at the plasma membrane in CUL3- or KCTD10-knockdown SKBR-3 cells as was seen in control cells (Fig 4A). Treating SKBR-3 cells with PHT-7.3 diminished the localization of CNKSR1 at the plasma membrane (Fig 4B), suggesting that CNKSR1 localizes at the plasma membrane through recognition of the plasmalemmal PI(4,5)P$_2$ when RhoB expression is low. In contrast, colocalization of CNKSR1 and RhoB at the plasma membrane was observed in KCTD10-depleted SKBR-3 cells even by the treatment with PHT-7.3 (Fig 4B). PHT-7.3 did not inhibit the interaction between CNKSR1 and RhoB (Q63L) in vitro (Fig 4C). As shown in Fig 2I, PI(4,5)P$_2$ did not inhibit the direct interaction between CNKSR1 and RhoB-GTP in vitro. Taken together, these data suggest that CNKSR1 interacts with RhoB-GTP at the plasma membrane in a PI(4,5)P$_2$-independent manner.

### Low expression of RhoB mRNA and high expression of CNKSR1 protein correlate with poor prognosis for HER2-positive breast cancer patients

Breast cancers can be classified based on their gene expression into five subtypes, namely, luminal-A (estrogen receptor (ER)$^+$, progesterone receptor (PR)$^+$, Ki-67$^{low}$, HER2$^-$), luminal-B (ER$^+$, PR$^+$, Ki-67$^{high}$, HER2$^{+/-}$), HER2-positive (ER$^-$, PR$^-$, HER2$^+$), basal (ER$^-$, PR$^-$, HER2$^-$, cytokeratin5/6$^+$, EGFR$^+$), and claudin-low (ER$^-$, PR$^-$, HER2$^-$, claudin$^{low}$) (Curtis et al, 2012; Gao et al, 2013; Pereira et al, 2016). Among the various breast cancer cell lines, we found that CNKSR1

control, was shown. **(D)** SKBR-3 cells were transfected with the indicated siRNAs. Cell lysates were prepared and subjected to pull-down with Rhotekin-conjugated beads. Cell lysates and the pull-downed samples were analyzed by Western blot. **(E)** Western blots of SKBR-3 lysates, 72 h post-transfection with the indicated siRNAs. Cells were treated with C3 transferase (1 μg/ml) for 24 h before preparation of cell lysates. pEGFR, phosphorylated EGFR (Tyr$^{1068}$); pHER2, phosphorylated HER2 (Tyr$^{1221/1222}$). The ratio of band intensities for pEGFR/EGFR or pHER2/HER2, normalized to control, was shown. **(F)** SKBR-3 cells were treated with indicated siRNAs for 48 h. Trypsinized cells (total $0.5 \times 10^5$ cells) were then replated and treated with the same siRNA. Cell number was counted 72 h after replating. Data are mean ± SEM from three independent experiments. **P < 0.01. **(G)** Western blots of cell lysates of SKBR-3 cells treated with MLN4924 (1 μM) for 24 h. The asterisk indicates neddylated-CUL3 (Nedd8-CUL3). pEGFR, phosphorylated EGFR (Tyr$^{1068}$); pHER2, phosphorylated HER2 (Tyr$^{1221/1222}$). The ratio of band intensities for pEGFR/EGFR or pHER2/HER2, normalized to control, was shown. **(H)** Scheme of EGFR and HER2 phosphorylation regulated by the CUL3/KCTD10/RhoB axis. The CUL3/KCTD10 E3 complex constitutively ubiquitinates RhoB leading to its degradation. In CUL3 or KCTD10 knockdown cells, accumulated RhoB-GTP reduced the phosphorylation of EGFR and HER2, resulting in the inhibition of cell proliferation.
Source data are available for this figure.

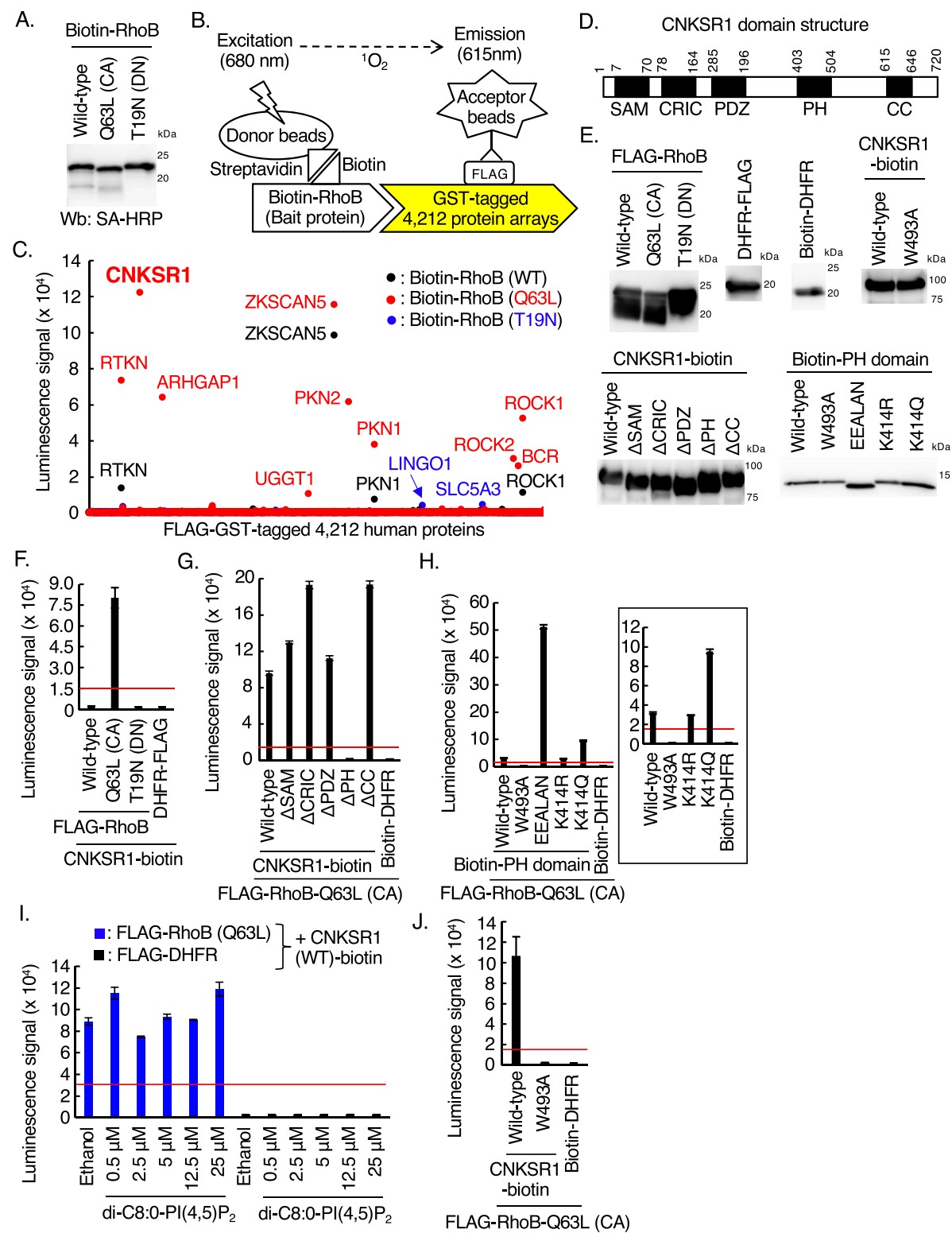

**Figure 2. Identification of connector enhancer of kinase suppressor of Ras1 (CNKSR1) as an interacting protein of RhoB-GTP using a cell-free–based human protein array.**
**(A)** Western blots of cell-free synthesized biotinylated RhoB (wild-type, Q63L, and T19N mutants) using the wheat cell-free system as bait proteins. CA, constitutive-active; DN, dominant-negative; SA-HRP, streptavidin-horseradish peroxidase. **(B)** Schematic diagram of the in vitro binding assay between bait proteins (wild-type, Q63L, or T19N mutants of RhoB) and FLAG-GST–tagged human proteins using AlphaScreen technology. **(C)** Results of in vitro high-throughput screen targeting 4,212 human proteins. See also Table S1. **(D)** The domain structure of human CNKSR1. The numbers in the domain structure represent the order of amino acids from the N-terminus. CNKSR1

was highly expressed in a HER2-positive breast cancer cell line, SKBR-3, and protein expression of CNKSR1 was low in MCF-7 (luminal-type), MDA-MB-231 (basal-type), and MDA-MB-453 (HER2-positive–type) cells (Fig 5A). We next examined the phosphorylation profile of EGFR and HER2 in those cell lines. The phosphorylation of neither EGFR nor HER2 was detected in MCF-7, MDA-MB-231, and MDA-MB-453 cells in the absence of exogenous EGF stimulation (Fig S2). These data suggest that the RhoB/CNKSR1 axis contributes to cell proliferative events in HER2/EGFR-double positive breast cancers. We thus examined if the expression level of RhoB correlates with the prognosis of human HER2/EGFR-double positive breast cancer patients using the Molecular Taxonomy of Breast Cancer International Consortium (METABRIC) database (Curtis et al, 2012). Although the expression of *EGFR* mRNA did not positively correlate with the prognosis of human breast cancer patients (Fig 5B, high: n = 974, low: n = 930), high mRNA expression of *EGFR* significantly correlated with poor prognosis for HER2-positive breast cancer patients (Fig 5C; *P* = 0.034, high: n = 198, low: n = 22). We also found that low mRNA expression of *RhoB* significantly correlated with poor prognosis for HER2-positive/EGFR[high] breast cancer patients (Fig 5D; *P* = 0.0053, high: n = 54, low: n = 30). In contrast, high mRNA expression of *RhoB* significantly correlated with poor prognosis for HER2-positive/EGFR[low] breast cancer patients (Fig 5E; *P* < 0.0001, high: n = 14, low: n = 96). Low mRNA expression of *RhoB* also significantly correlated with poor prognosis for HER2[high] and HER2[high]/CNKSR1[high] breast cancer patients (Fig 5F; *P* = 0.00089, high: n = 172, low: n = 304, Fig 5G; *P* = 0.021, high: n = 92, low: n = 146, respectively). Taken together, these data suggested that RhoB might function as a tumor suppressor in HER2/EGFR-double positive breast cancers.

We next examined the protein expression of CNKSR1 in HER2-positive human breast cancer tissues. In total, 20 individual primary tumor tissues from patients with HER2-positive breast cancer were subjected to immunohistochemistry. CNKSR1 was highly expressed in 80% of HER2-positive breast cancer tissues (Table 1 and Fig 5H). Of note, all tested primary lesions of HER2-positive breast cancers with metastatic recurrence were CNKSR1-positive (Table 1). High protein expression of CNKSR1 was associated with poor prognosis for HER2-positive breast cancer patients (Fig 5I). In a HER2/EGFR-double positive breast cancer cell line, SKBR-3, the overexpression of Myc-tagged or non-tagged CNKSR1 increased the phosphorylation of both EGFR and HER2, as well as cell proliferation (Fig 6A–D). These data suggest that high CNKSR1 expression enhances cell proliferative signaling in HER2-positive breast cancers.

### A plasmalemmal EGFR phosphatase, PTPRH, is activated by the accumulation of RhoB-GTP through its release from CNKSR1

We finally elucidated the molecular mechanisms underlying the CNKSR1-mediated phosphorylation of EGFR and HER2, which are regulated by the CUL3/KCTD10/RhoB axis. We noticed that the PH

domain and a Trp[493] residue in the PH domain were required for CNKSR1-mediated EGFR/HER2 phosphorylation and cell proliferation, although the protein expression of RhoB remained low (Fig 6A–D). These data suggest that CNKSR1 has a capacity to enhance EGFR and HER2 phosphorylation in a Trp[493] residue in the PH domain-dependent manner, and the ability does not require RhoB-GTP. Knockdown of CUL3 or KCTD10 reduced the phosphorylation of both EGFR and HER2 in CNKSR1 (W493A) mutant-expressing SKBR-3 cells, as seen in control and wild-type CNKSR1-expressing SKBR-3 cells (Fig 6E). These data suggest that a CNKSR1 (W493A) mutant functions as a loss-of-function mutant in the regulation of EGFR and HER2 phosphorylation. Since phosphorylation is balanced by kinase and phosphatase reactions, kinase activities of EGFR and/or HER2 are inactivated, or phosphatase activities for EGFR and/or HER2 are activated, upon CUL3- or KCTD10-depletion. Stimulation of SKBR-3 cells through the addition of EGF exogenously induced the phosphorylation of EGFR, HER2, Akt, and ERK-1/2 in CUL3- or KCTD10-depleted SKBR-3 cells, suggesting that kinase activities of EGFR and HER2 were enzymatically active (Fig 6F). The localization and mobility of EGFR were not affected by the knockdown of CUL3 or KCTD10 (Murakami et al, 2019) (Fig S3A and B). We thus speculated that EGFR/HER2 phosphatases and RhoB-GTP exclusively interact with CNKSR1 at the plasma membrane (Fig 6G).

To identify the EGFR/HER2 phosphatases that directly interact with CNKSR1 in a Trp[493] residue–dependent manner, we prepared a human phosphatase array that consists of 171 FLAG-GST–tagged human phosphatases synthesized by the wheat cell-free protein synthesis system (Table S2 and Fig 7A). Biotinylated CNKSR1 (wild-type, W493A mutant) and RhoB (Q63L) recombinant proteins (bait proteins) were also produced by the wheat cell-free protein synthesis system and reacted with the human phosphatase array. The AlphaScreen assay revealed 15 of the CNKSR1-bound phosphatases (Fig 7A). We then silenced the each of corresponding 15 genes in KCTD10-depleted SKBR-3 cells and found that the knockdown of PTPRH partially restored the decreased phosphorylation of EGFR in KCTD10-depleted SKBR-3 cells (Figs S4 and 7B). Knockdown of any CNKSR1-interacting phosphatases including PTPRH did not restore the decreased level of HER2 phosphorylation in KCTD10-depleted SKBR-3 cells (Figs S4 and 7B), suggesting the redundancy of several CNKSR1-interacting phosphatases in the regulation of HER2 phosphorylation. PTPRH knockdown in KCTD10-depleted SKBR-3 cells significantly restored the inhibition of cell proliferation (Fig 7C). PTPRH (also known as stomach cancer-associated protein-tyrosine phosphatase; SAP-1) is a transmembrane-type phosphatase that possesses a catalytic domain in its cytosolic region and fibronectin type III-like repeats in its extracellular region containing multiple N-glycosylation sites (Matozaki et al, 1994). PTPRH directly dephosphorylates EGFR and suppresses its downstream signaling pathways in HEK293 cells (Yao et al, 2017). We confirmed that the

---

possesses SAM, CRIC, PDZ, PH, and CC domain. **(E)** Western blots of cell-free synthesized biotinylated and FLAG-tagged proteins. Δ, deletion form of each domain. FLAG-tagged proteins and biotinylated proteins were detected using anti-FLAG antibody and streptavidin-horseradish peroxidase, respectively. **(F, G, H, I, J)** In vitro binding assay for determination of the biotinylated proteins/FLAG-tagged proteins interaction using AlphaScreen technology. Proteins synthesized by wheat germ extracts (Fig 2E) were subjected to AlphaScreen as indicated. DHFR was used as a negative control. The red lines indicate the threshold of interactions (10 times the luminescence signal for CNKSR1 or RhoB-Q63L/DHFR). Data from three independent experiments are expressed as the means ± SEM. Δ, deletion form of each domain. In Fig 2H, a graph without data obtained from a EEALAN mutant (K414E/P416E/R423A/R425L/R426A/K478N) is shown. In Fig 2I, the protein mixtures were incubated with di-C8:0-PI(4,5)P$_2$ for 1 h. Source data are available for this figure.

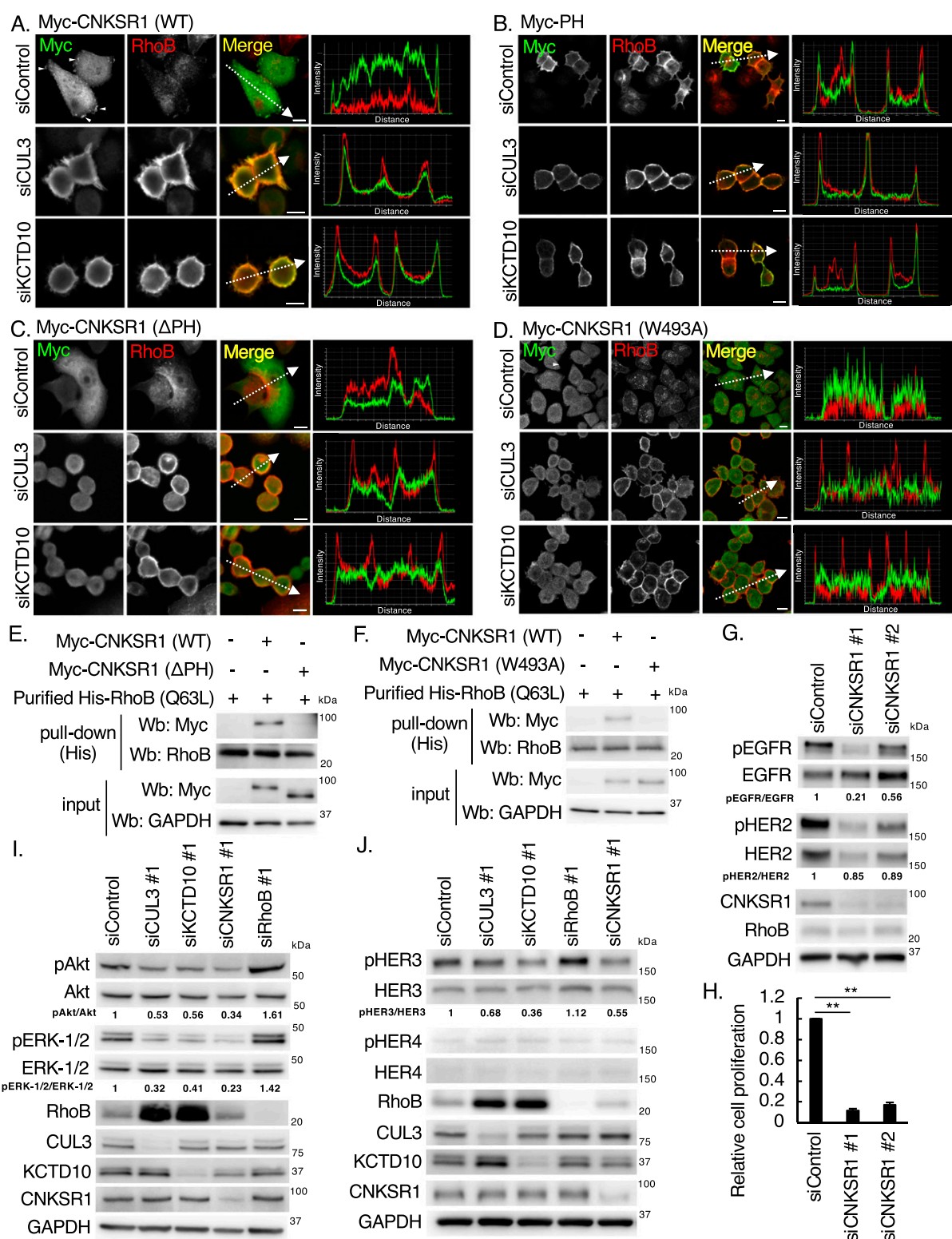

**Figure 3. The interaction of connector enhancer of kinase suppressor of Ras1 (CNKSR1) with RhoB-GTP and effects of CNKSR1 knockdown on signal transduction in SKBR-3 cells.**
**(A, B, C, D)** Confocal images of SKBR-3 cells treated with control siRNA, CUL3 siRNA #1 or KCTD10 siRNA #1 for 72 h. 48 h after infection of the Myc-CNKSR1 (wild-type, ΔPH, or W493A) or Myc-PH-carrying lentivirus, cells were fixed, permeabilized, and stained for Myc and RhoB. Membrane ruffles were indicated by arrowheads. A fluorescence intensity profile along the arrow in the image is shown in right panel. Bars; 10 μm. **(E, F)** SKBR-3 cells were transfected with Myc-CNKSR1 (wild-type, ΔPH, or W493A) vectors. Cell lysates were prepared and subjected to pull-down with purified His-RhoB (Q63L) that mimics RhoB-GTP. Cell lysates and the pull-downed samples were

cytosolic region of PTPRH directly interacts with CNKSR1 but not RhoB (Q63L) and that a Trp[493] residue of CNKSR1 is responsible for the interaction with both PTPRH and RhoB (Q63L) in vitro (Fig 7D and E). The overexpression of wild-type PTPRH-HA, but not the phosphatase activity-dead mutant C1020S (Takada et al, 2002), reduced the phosphorylation of EGFR (Fig 7F). The phosphorylation of HER2 was not remarkably decreased by overexpression of wild-type PTPRH-HA (Fig 7F). Notably, overexpression of PTPRH-HA (C1020S) relieved the decrease in EGFR phosphorylation in KCTD10-depleted cells (Fig 7F). In parallel with changes in the phosphorylation of EGFR, the overexpression of wild-type PTPRH-HA, but not the C1020S mutant, significantly suppressed the proliferation of SKBR-3 cells (Fig 7G). The overexpression of PTPRH-HA (C1020S) attenuated the inhibition of cell proliferation induced by KCTD10 knockdown (Fig 7G). These data suggested that PTPRH is activated by KCTD10 knockdown, followed by the accumulation of RhoB-GTP. Colocalization of PTPRH and EGFR was observed at the plasma membrane in both control and KCTD10-depleted SKBR-3 cells (Fig 7H). Knockdown of KCTD10, which causes the accumulation of RhoB-GTP, reduced the interaction between PTPRH and CNKSR1 in SKBR-3 cells (Fig 7I). The addition of purified His-tagged RhoB-GTP significantly reduced the enhanced luminescence signals between CNKSR1 and the cytosolic region of PTPRH in vitro (Figs 7J and S1). These data suggest that RhoB-GTP and PTPRH exclusively interact with CNKSR1.

We then sought to identify the responsible regions of PTPRH for the interaction with CNKSR1. For this aim, we generated seven deletion mutants of PTPRH-cyto, which delete 50 or 39 sequential amino acids in the cytosolic region of PTPRH (Fig 8A and B). As shown in Fig 8C, the in vitro binding assay by AlphaScreen indicated that luminescence signals between CNKSR1 and the cytosolic region of PTPRH reduced to below a threshold in the del-1, del-3, del-4, and del-5 mutants. These data suggest that CNKSR1 may recognize a specific three-dimensional structure of the overall cytosolic region of PTPRH. Collectively, we concluded that the constitutive degradation of RhoB by the CUL3/KCTD10 E3 complex enables CNKSR1 to interact with PTPRH, leading to the inactivation of EGFR phosphatase activities of PTPRH (Fig 8D). In CUL3- or KCTD10-knockdown cells, the accumulated RhoB-GTP at the plasma membrane interacts with CNKSR1, followed by the release and activation of PTPRH, leading to the dephosphorylation of EGFR (Fig 8E).

# Discussion

Our result indicated that the depletion of CUL3 or KCTD10 caused the accumulation of RhoB protein in HER2-positive breast cancer cells. Physiologically, a microRNA, miR-101, induced by hypoxia targets to the 3′ untranslated region of CUL3, which down-regulates protein expression of CUL3 in human umbilical vein endothelial cells resulting in the up-regulation of VEGF and promotion of angiogenesis both in vitro and in vivo (Kim et al, 2014). The hydrogen peroxide induces expression of miR-455 and miR-601, which down-regulates protein expression of CUL3 in osteoblasts and retinal pigment epithelium cells, respectively (Xu et al, 2017; Chen et al, 2019). The miR-592 may down-regulate KCTD10 expression during the development of congenital heart diseases (Pang et al, 2019). It is likely that expression of CUL3 and KCTD10 would be reduced by microRNAs also in HER2-positive breast cancer cells under specific patho-physiological conditions.

We noticed that the depletion of CUL3 or KCTD10 increased the protein expression of EGFR and reduced the protein expression of HER2 in SKBR-3 cells (Figs 1B and 6F). Knockdown of CNKSR1 reduced the protein expression of HER2 in SKBR-3 cells (Fig 3G). Knockdown of neither CUL3, KCTD10, nor CNKSR1 significantly affected the level of mRNA expression of EGFR and HER2 (Fig S5A and B). Because CUL3 is essential for trafficking and degradation of EGFR (Huotari et al, 2012; Gschweitl et al, 2016), these data suggested that CUL3, KCTD10, and CNKSR1 may contribute to protein turnover or/and mRNA translation of EGFR and HER2 in SKBR-3 cells.

CNKSR1 functions as an oncogene and drives anchorage-independent cell growth through RAF/MEK/ERK phosphorylation in K-Ras–mutated lung and colorectal cancers (Indarte et al, 2019). CNKSR1 can accelerate RAF/MEK/ERK signaling via positive feedback effects through its plasmalemmal localization (Fischer et al, 2017). The phosphorylation of Akt is positively regulated by CNKSR1 in basal-type breast cancer MDA-MB-231 cells (Fritz et al, 2010). In contrast, PTPRH serves as a tumor suppressor. The overexpression of PTPRH suppresses growth factor-induced ERK activation and colony formation (Noguchi et al, 2001). Furthermore, enforced expression of PTPRH reduces the phosphorylation of Akt and GSK-3$\alpha$/$\beta$, which induces apoptosis (Takada et al, 2002). Clinically, the protein abundance of PTPRH is inversely related to the aggressiveness of human hepatocellular carcinoma patients, and enforced expression of PTPRH inhibits the migration and growth of human hepatocellular carcinoma cells (Nagano et al, 2003). Whole genome sequencing of tumors derived from a mouse model of breast cancer, MMTV-PyMT (Guy et al, 1992), identified a highly conserved mutation in PTPRH, which increases the phosphorylation of EGFR (Rennhack et al, 2019). Our present results suggest that CNKSR1 serves as an oncoprotein by inactivating PTPRH, a tumor-suppressive EGFR phosphatase, in HER2-positive breast cancer cells. RhoB acts as both an oncogene and a tumor suppressor in a context-dependent manner (Ju & Gilkes, 2018). The notion is supported our expression analysis using METABRIC database because the correlation between *RhoB* mRNA expression and prognosis of HER2-positive breast cancer patients were reversed dependent on expression level of *EGFR* mRNA (Fig 5D and E). For HER2/EGFR double-positive breast cancer cells, our present study suggests that RhoB functions as a

---

analyzed by Western blot. See also Fig S1. **(G)** Western blots of SKBR-3 lysates, 72 h post-transfection with the indicated siRNAs for CNKSR1. pEGFR, phosphorylated EGFR (Tyr[1068]); pHER2, phosphorylated HER2 (Tyr[1221/1222]). The ratio of band intensities for pEGFR/EGFR or pHER2/HER2, normalized to control, was shown. **(H)** SKBR-3 cells were treated with indicated siRNAs for 48 h. Trypsinized cells (total 0.5 × 10^5 cells) were then replated and treated with the same siRNA. Cell number was counted 72 h after replating. Data are mean ± SEM from three independent experiments. **$P$ < 0.01. **(I, J)** Western blots of SKBR-3 lysates, 72 h post-transfection with the indicated siRNAs. pAkt, phosphorylated Akt (Ser[473]); pERK-1/2, phosphorylated ERK-1/2 (Thr[202]/Tyr[204]); pHER3, phosphorylated HER3 (Tyr[1222]); pHER4, phosphorylated HER4 (Tyr[1284]). The ratio of band intensities for pAkt/Akt, pERK-1/2/ERK1/2, or pHER3/HER3, normalized to control, was shown.
Source data are available for this figure.

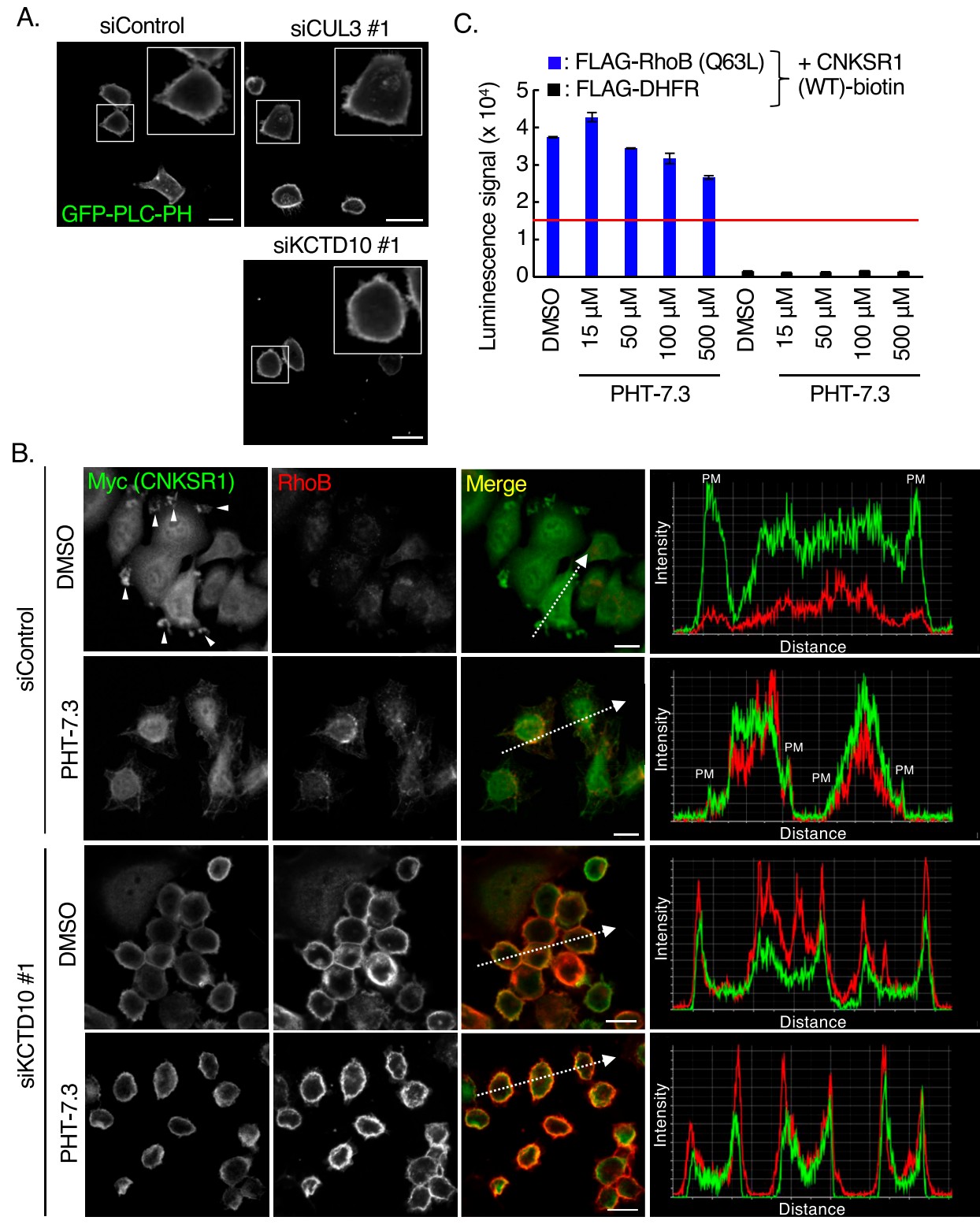

**Figure 4.   Relationships between PI(4,5)P2 and RhoB-GTP/connector enhancer of kinase suppressor of Ras1 (CNKSR1) interaction at the plasma membrane.**
**(A)** Confocal images of SKBR-3 cells treated with control siRNA, CUL3 siRNA #1 or KCTD10 siRNA #1 for 72 h. 48 h after transfection of the GFP-PLC-PH vector (a biosensor for PI[4,5]P2), cells were fixed. Bars; 10 $\mu$m. **(B)** Confocal images of SKBR-3 cells treated with control siRNA or KCTD10 siRNA #1 for 72 h. 48 h after infection of the Myc-CNKSR1–carrying lentivirus, cells were fixed, permeabilized, and stained for Myc and RhoB. Cells were treated with PHT-7.3 (50 $\mu$M) for 24 h before the fixation. Membrane ruffles were indicated by arrowheads. A fluorescence intensity profile along the arrow in the image is shown in right panel. PM, plasma membrane. Bars; 10 $\mu$m. **(C)** In vitro binding assay for determination of the biotinylated CNKSR1/FLAG-tagged proteins interaction using AlphaScreen technology. Proteins synthesized by wheat germ extracts (Fig 2E) were subjected to AlphaScreen as indicated. DHFR was used as a negative control. The red lines indicate the threshold of interactions (10 times the luminescence signal for CNKSR1/DHFR). Data from three independent experiments are expressed as the means ± SEM. The protein mixtures were incubated with PHT-7.3 for 1 h.

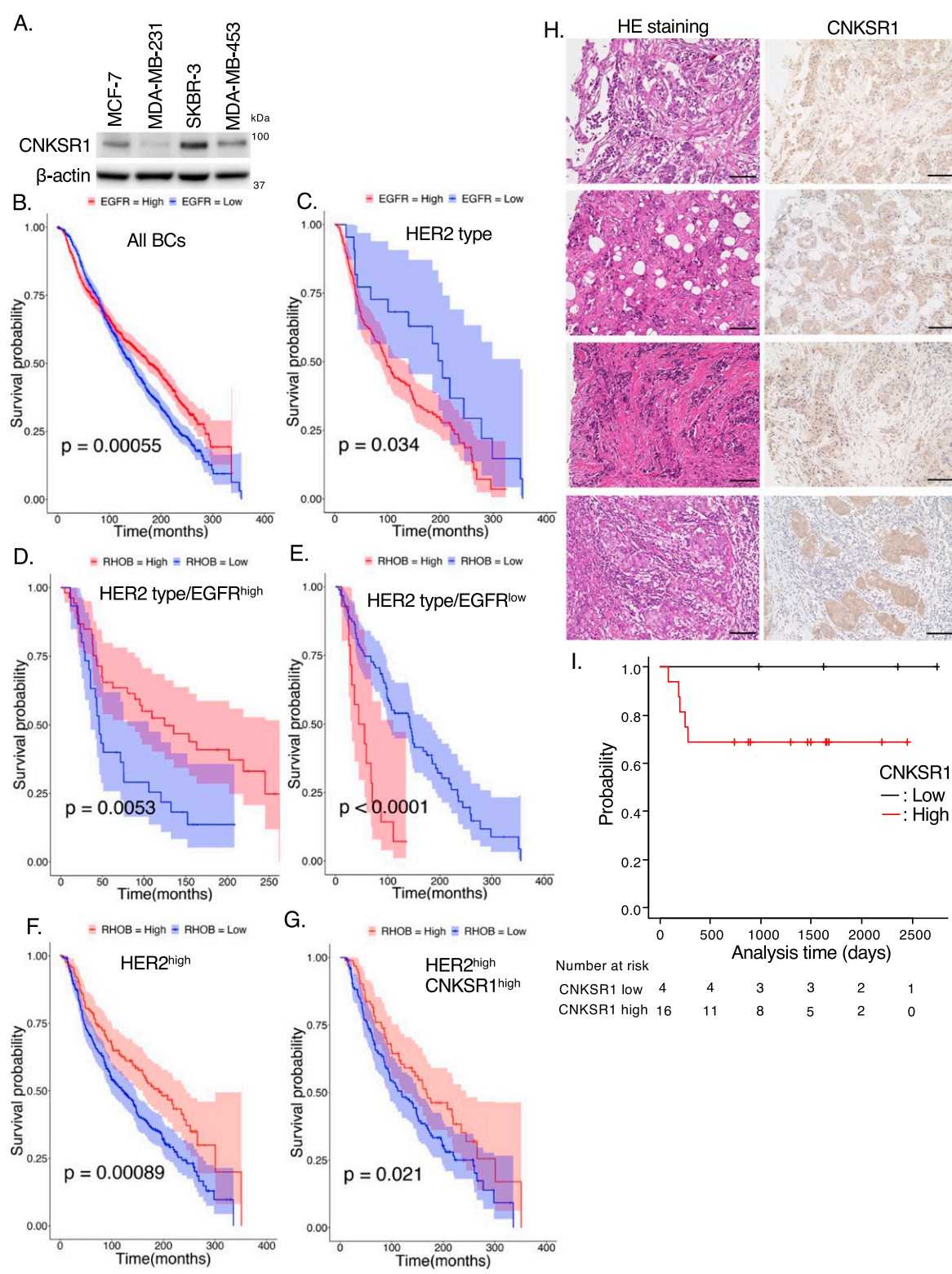

**Figure 5. Clinical significances of RhoB and connector enhancer of kinase suppressor of Ras1 (CNKSR1) in human HER2-positive breast cancers. (A)** Western blots of human breast cancer cell lines. MCF-7 cells (luminal type), MDA-MB-231 cells (basal type), SKBR-3 cells, and MDA-MB-453 cells (HER2-positive). **(B, C, D, E, F, G)** Kaplan–Meier plot with epidermal growth factor receptor (EGFR) (B, C) or RhoB (D, E, F, G) expression profiles in Molecular Taxonomy of Breast Cancer International Consortium cohort using overall survival status. Shading along the curve showed 95% confidential interval. **(B)** Analysis with all samples showed that high expression of EGFR was not correlated to poor prognosis (B; high: n = 974, low: n = 930). **(C)** In HER2-positive subtype, high expression of EGFR was correlated to poor prognosis (C; high: n = 198, low: n = 22). **(D)** In HER2-positive/EGFR^high type, low mRNA expression of RhoB was correlated to poor prognosis (D; high: n = 54, low: n = 30). **(E)** In HER2-positive/

**Table 1. Expression profile of connector enhancer of kinase suppressor of Ras1 (CNKSR1) in human HER2-positive breast cancer tissues.**

| | | CNKSR1 low (n = 4) | CNKSR1 high (n = 16) |
|---|---|---|---|
| Age | Average | 67.75 (range 58–88) | 61.50 (range 36–90) |
| Level of invasion | T1 | 0/8 (0%) | 8/8 (100%) |
| | T2 | 4/9 (44%) | 5/9 (56%) |
| | T3 | 0/1 (0%) | 1/1 (100%) |
| | T4 | 0/2 (0%) | 2/2 (100%) |
| Lymph node metastasis | N0 | 2/10 (20%) | 8/10 (80%) |
| | N1 | 1/3 (33%) | 2/3 (67%) |
| | N2 | 1/3 (33%) | 2/3 (67%) |
| | N3 | 0/4 (0%) | 4/4 (100%) |
| Stage | I | 0/7 (0%) | 7/7 (100%) |
| | II | 3/5 (60%) | 2/5 (40%) |
| | III | 1/8 (12.5%) | 7/8 (87.5%) |
| Distant metastasis | Absent | 4/15 (26.7%) | 11/15 (73.3%) |
| | Present | 0/5 (0%) | 5/5 (100%) |
| Ki-67 | Average | 49.75 (range 20–74) | 55.75 (range 30–90) |

Total 20 of human HER2-positive breast cancer tissues were subjected to immunohistochemistry for CNKSR1. The number of human breast cancer tissues with CNKSR1-positive or CNKSR1-negative was counted. Breast cancer cells were defined as CNKSR1-positive when the signals for the CNKSR1-DAB staining were positive above the threshold 10 after subtraction.

tumor suppressor because its constitutive degradation by the CUL3/KCTD10 E3 complex is essential for cell proliferation of SKBR-3 cells.

In this study, we propose that CNKSR1 forms a complex with PTPRH to inactivate its EGFR-targeting protein tyrosine phosphatase activity and that released PTPRH from CNKSR1, through the formation of a RhoB-GTP/CNKSR1 complex, exerts tyrosine phosphatase activity on EGFR (Fig 8C and D). RhoB has the ability to interact with and activate a specific serine/threonine phosphatase, protein phosphatase 2A, through the recruitment of its regulatory subunit B55, resulting in Akt dephosphorylation (Bousquet et al, 2016). RhoB might activate tyrosine and serine/threonine phosphatases through distinct molecular pathways. The release of auto-inhibition through the binding of an N-SH2 domain of the protein tyrosine phosphatase PTPN11 (alias SHP2) to a phosphorylated tyrosine residue is a familiar mechanism of SHP2 activation (Frankson et al, 2017). Regarding to PTPRH, we showed that CNKSR1 could recognize a specific spatial structure of the cytosolic region of PTPRH (Fig 8A–C). The deletion of amino acid sequences including a cysteine residue (Cys$^{1020}$), which is essential for phosphatase activities, resulted in the loss of capacity to interact with CNKSR1 (the del-5 mutant, Fig 8A–C). In the PTPRH/CNKSR1 complex, it is likely that the phosphatase-active site of PTPRH is masked through the interaction with CNKSR1, which might inactivate PTPRH phosphatase activity. Upon accumulation of RhoB-GTP, CNKSR1 forms a complex with RhoB-GTP, resulting in the release and activation of PTPRH. The structural analysis of the RhoB-GTP/CNKSR1 complex as well as the PTPRH/CNKSR1 complex by X-ray crystallography or cryo-electron microscopy would be required for further discussion of their conformational regulations. Here, we identified 15 CNKSR1-bound phosphatases from a human phosphatase array using in vitro AlphaScreen technology (Fig 7A and Table S2). It is likely that the phosphatase activities of those CNKSR1-bound phosphatases could be controlled through a "non-autoinhibitory" direct interaction with CNKSR1 in other cancer cell lines, instead of SKBR-3 cells. PTPRH possesses multiple N-glycosylation sites in its extracellular regions (Matozaki et al, 1994). Another glycosylated protein tyrosine phosphatase receptor, PTPRC (alias CD45), reduces accessibility of large ligands (e.g., dead cells) to phagocytic receptors through the formation of extracellular "glycocalyx pickets" (Freeman et al, 2016; Ostrowski et al, 2016). PTPRC also limits mobility of the phagocytic receptors through the formation of cortical actin fences in cytosol (Freeman et al, 2016; Ostrowski et al, 2016). The glycosylated extracellular regions of PTPRH would function as glycocalyx

EGFR$^{low}$ type, high mRNA expression of RhoB was correlated to poor prognosis (E; high: n = 14, low: n = 96). **(F, G)** In both HER2$^{high}$ and HER2$^{high}$/CNKSR1$^{high}$ breast cancers, low mRNA expression of RhoB was correlated to poor prognosis (F; High: n = 172, Low: n = 304, G; high: n = 92, low: n = 146, respectively). **(H)** Expression of CNKSR1 protein in human HER2-positive breast cancer tissues (n = 20). Representative images of the hematoxylin and eosin staining and immunohistochemical staining for CNKSR1 in primary lesions of HER2-positive breast cancers with metastatic recurrence were shown. The neighboring sections of the human HER2-positive breast cancer tissues were subjected to hematoxylin and eosin staining and immunohistochemistry. Bars; 100 µm. See also Table 1. **(I)** The Kaplan–Meier plot with protein expression profiles of CNKSR1 in human HER2-positive breast cancer using end point disease-free survival status. The high expression of CNKSR1 was correlated to poor prognosis for disease-free survival (high: n = 16, low: n = 4).
Source data are available for this figure.

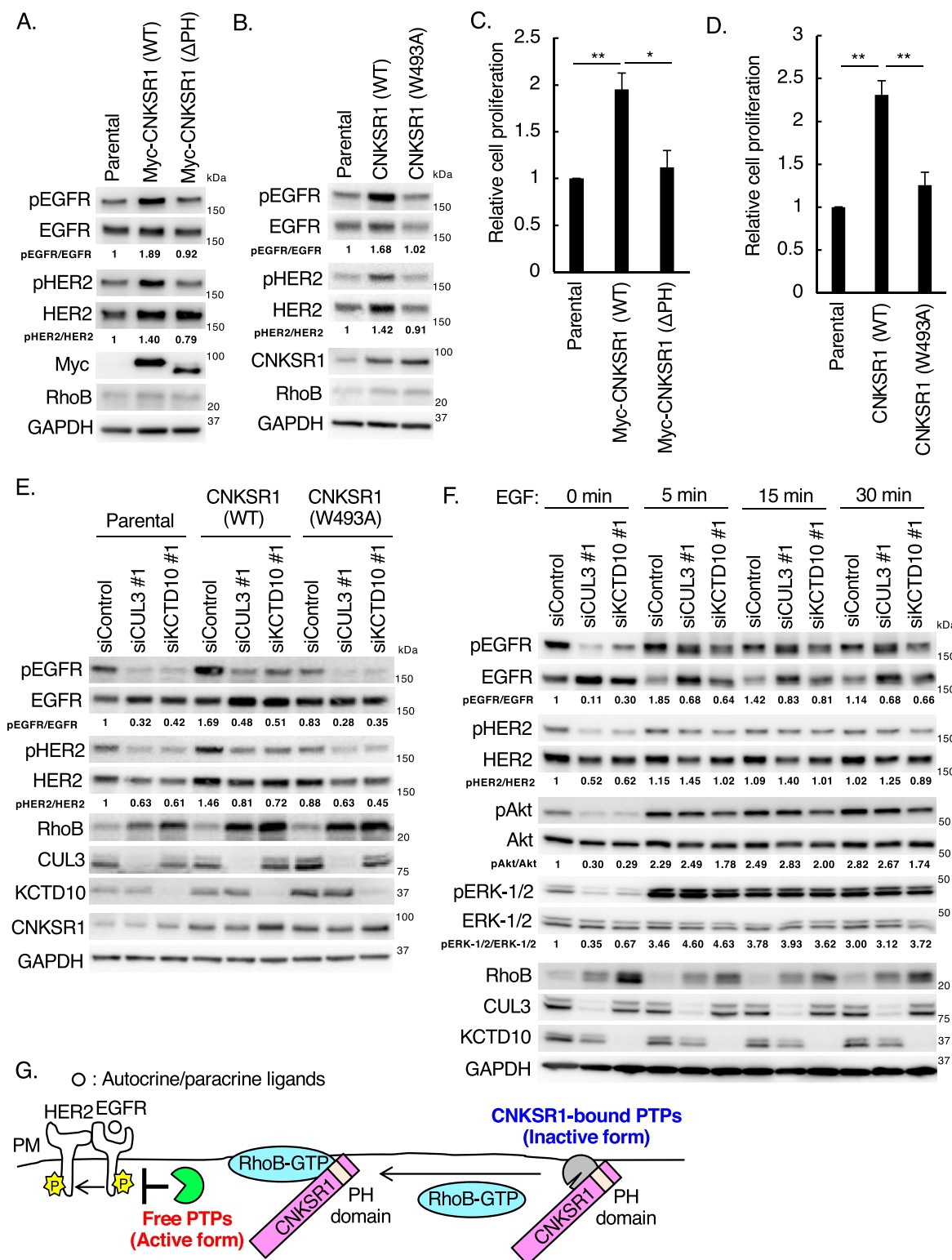

**Figure 6.   Effects of connector enhancer of kinase suppressor of Ras1 (CNKSR1) overexpression on epidermal growth factor receptor (EGFR)/HER2 signaling in SKBR-3 cells.**
**(A, B)** Western blots of SKBR-3 lysates stably expressing Myc-CNKSR1 (wild-type or ΔPH) or non-tagged CNKSR1 (wild-type or W493A). pEGFR, phosphorylated EGFR (Tyr[1068]); pHER2, phosphorylated HER2 (Tyr[1221/1222]). The ratio of band intensities for pEGFR/EGFR or pHER2/HER2, normalized to control, was shown. **(C, D)** Trypsinized SKBR-3 cells stably expressing Myc-CNKSR1 (wild-type or ΔPH) or non-tagged CNKSR1 (wild-type or W493A) (total $0.5 \times 10^5$ cells) were plated. Cell number was counted 72 h after replating. Data are mean ± SEM from three independent experiments. *$P < 0.05$, **$P < 0.01$. **(E)** SKBR-3 cells stably expressing non-tagged CNKSR1 (wild-type or W493A) were

pickets like PTPRC. The level and pattern of PTPRH glycosylation may also regulate the accessibility of EGFR to PTPRH at the plasma membrane.

In SKBR-3 cells, PTPRH partially contributed to the decrease in EGFR phosphorylation in KCTD10-knockdown cells (Fig 7B). The high expression of *PTPRH* mRNA was significantly correlated with high *EGFR* mRNA expression in 13 HER2-positive breast cancer cell lines (Fig S6), suggesting the functional significance of PTPRH in the dephosphorylation of EGFR in HER2-positive breast cancer cells. As shown in Fig S7, EGFR phosphorylation was detected in neither A431 cells (a human epidermoid carcinoma cell line; PTPRH[high]), HepG2 cells (a human hepatocellular carcinoma cell line; PTPRH[low]), nor MIA-PaCa-2 cells (a human pancreatic carcinoma cell line; PTPRH[low]) in the steady-state condition. These data suggest that EGFR is not activated by endogenously produced ligands in those cancer cells. Knockdown of CUL3, but not KCTD10, increased the protein expression of RhoB in A431 cells (Fig S7A), suggesting the contribution of other BTBPs for RhoB degradation in A431 cells. In HepG2 cells, CUL3 or KCTD10 knockdown caused the accumulation of RhoB protein as was seen in SKBR-3 cells (Fig S7B). In contrast, RhoB protein was accumulated by knockdown of neither CUL3 nor KCTD10 in MIA-PaCa-2 cells (Fig S7C). Taken together, it is likely that the regulation of EGFR phosphorylation through PTPRH activation and RhoB degradation by the CUL3/KCTD10 E3 complex is conserved in HER2/EGFR-double positive breast cancer cells. PTPRH is expressed in normal epithelial cells of the small intestine and stomach in mice (Sadakata et al, 2009; Murata et al, 2015). The roles of CUL3/KCTD10 E3 complex and CNKSR1 in the PTPRH regulation in the normal epithelial cells would be further investigated in future.

Depletion of PTPRH in KCTD10 knockdown-SKBR-3 cells did not restore the decrease of HER2 phosphorylation and partially recovered the decrease in EGFR phosphorylation (Fig 7B). These results suggest the contribution of other phosphatases to HER2 and EGFR dephosphorylation in KCTD10-depleted cells (Fig 8E). Several phosphatases could redundantly function in dephosphorylation of HER2 and EGFR with PTPRH. From a drug-discovery standpoint because inhibition of the PTPRH/CNKSR1 complex releases PPTRH leading to its activation, inhibitors of this PTPRH/CNKSR1 interaction could represent novel drugs for HER2-positive breast cancers.

# Materials and Methods

### Antibodies

The following antibodies were purchased from the indicated manufacturers: rabbit anti-EGFR antibody (D38B1, dilution 1:1,000; Cell Signaling Technology), rabbit anti-phosphorylated EGFR antibody (Tyr[1068]) (D7A5, dilution 1:1,000; Cell Signaling Technology), rabbit anti-HER2 antibody (D8F12, dilution 1:1,000; Cell Signaling Technology), rabbit anti-phosphorylated HER2 antibody (Tyr[1221/1222]) (6B12, dilution 1:1,000; Cell Signaling Technology), rabbit anti-HER3 antibody (D22C5, dilution 1:1,000; Cell Signaling Technology), rabbit anti-phosphorylated HER3 antibody (Tyr[1222]) (50C2, dilution 1:1,000; Cell Signaling Technology), rabbit anti-HER4 antibody (111B2, dilution 1:1,000; Cell Signaling Technology), rabbit anti-phosphorylated HER4 antibody (Tyr[1284]) (21A9, dilution 1:1,000; Cell Signaling Technology), rabbit anti-RhoA antibody (67B9, dilution 1:1,000; Cell Signaling Technology), rabbit anti-RhoB antibody (D1J9V, dilution 1:1,000; Cell Signaling Technology), rabbit anti-RhoC antibody (D40E4, dilution 1:1,000; Cell Signaling Technology), mouse anti-CUL3 antibody (CUL3-9, dilution 1:1,000; Sigma-Aldrich), rabbit anti-KCTD10 antibody (HPA014273, dilution 1:1,000; Sigma-Aldrich), rabbit anti-CNKSR1 antibody (10885-1-AP, dilution 1:1,000 for Western blotting, 1:200 for immunohistochemistry; Proteintech), rabbit anti-Akt antibody (9272, dilution 1:1,000; Cell Signaling Technology), rabbit anti-phosphorylated Akt antibody (Ser[473]) (9271, dilution 1:1,000; Cell Signaling Technology), rabbit anti-ERK-1/2 antibody (137F5, dilution 1:1,000; Cell Signaling Technology), rabbit anti-phosphorylated ERK-1/2 antibody (Thr[202]/Tyr[204]) (9101, dilution 1:1,000; Cell Signaling Technology), mouse anti-$\beta$ actin antibody (6D1, dilution 1:1,000; Wako), rabbit anti-PTPRH antibody (ab231767, dilution 1:1,000; Abcam), mouse anti-HA antibody (F-7, dilution 1:1,000; Santa Cruz), mouse anti-GAPDH antibody (5A12, dilution 1:6,000; Wako), mouse anti-Myc antibody (9E10, dilution 1:1,000; Santa Cruz), mouse anti-FLAG antibody (M2, dilution 1:1,000; Sigma-Aldrich), goat Cy3-conjugated anti-rabbit IgG antibody (A10520, dilution 1:2,000; Molecular Probes), goat Alexa 488–conjugated anti-mouse IgG antibody (A11001, dilution 1:2,000; Molecular Probes), HRP-conjugated anti-mouse IgG antibody (W4021, dilution 1:2,000; Promega), and HRP-conjugated anti-rabbit IgG antibody (W4011, dilution 1:2,000; Promega).

### Plasmids

The RhoB gene was amplified from a vector containing the human RhoB cDNA (Murakami et al, 2019) using the following pairs of primers: 5'-ATGGCGGCCATCCGCAAGAA-3' (RhoB sense primer) and 5'-TCATAGCACCTTGCAGCAGT-3' (RhoB antisense primer). The PCR products were introduced into the blunt ends of the pEU-N-bls vector or the pEU-N-FLAG vector (king gifts from Ms Yuki Tanaka, Ehime University). The constitutive active RhoB mutant (Q63L) and dominant negative RhoB mutant (T19N) were generated with the following pairs of primers: 5'-GACACGGCGGGCCTGGAGGACTACG-3' (Q63L sense primer), 5'-CCACAGCGCCAGCTCCACCTGCTTG-3' (Q63L antisense primer), 5'-GCGTGTGGCAAGAACTGCCTGCTGA-3' (T19N sense primer), 5'-GCCGTCGCCCACCACCACCAGCTTC-3' (T19N antisense primer). The His-RhoB (Q63L) gene was amplified from a FLAG-RhoB (Q63L)

transfected with the indicated siRNAs. Cell lysates were prepared at 72 h post-siRNA transfection and subjected to Western blots. pEGFR, phosphorylated EGFR (Tyr[1068]); pHER2, phosphorylated HER2 (Tyr[1221/1222]). The ratio of band intensities for pEGFR/EGFR or pHER2/HER2, normalized to control, was shown. **(F)** SKBR-3 cells were transfected with the indicated siRNAs. At 72 h post-siRNA transfection, serum-starved SKBR-3 cells were treated with EGF (100 ng/ml) for the indicated time. Cell lysates were then prepared and subjected to Western blots. pEGFR, phosphorylated EGFR (Tyr[1068]); pHER2, phosphorylated HER2 (Tyr[1221/1222]); pAkt, phosphorylated Akt (Ser[473]); pERK-1/2, phosphorylated ERK-1/2 (Thr[202]/Tyr[204]). The ratio of band intensities for pEGFR/EGFR, pHER2/HER2, pAkt/Akt, or pERK-1/2/ERK1/2, normalized to control, was shown. **(G)** Scheme of existence of EGFR phosphatases regulated by the RhoB-GTP/CNKSR1 axis. PM, plasma membrane. Source data are available for this figure.

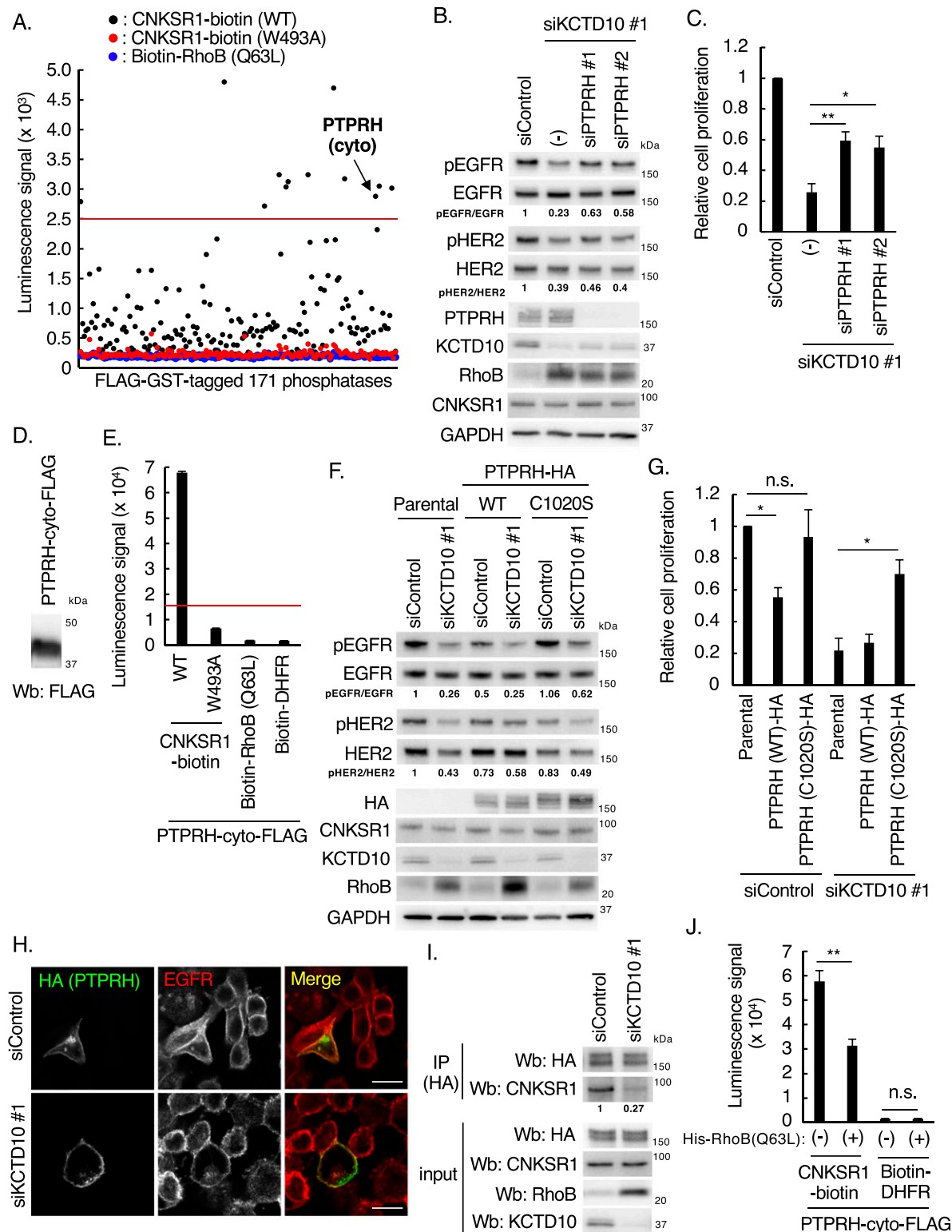

**Figure 7. Identification of PTPRH as an interacting protein of connector enhancer of kinase suppressor of Ras1 (CNKSR1) using a cell-free–based human phosphatase array.**

**(A)** Results of in vitro high-throughput screen targeting 171 human phosphatases. See also Table S2. The red line indicates the threshold of interactions (10 times the luminescence signal for CNKSR1-biotin/FLAG-DHFR). PTPRH (cyto), cytosolic region of PTPRH. **(B)** Western blots of SKBR-3 lysates, 72 h post-transfection with the indicated siRNAs. pEGFR, phosphorylated EGFR (Tyr[1068]); pHER2, phosphorylated HER2 (Tyr[1221/1222]). The ratio of band intensities for pEGFR/EGFR or pHER2/HER2, normalized to control, was shown. **(C)** SKBR-3 cells were treated with indicated siRNAs for 48 h. Trypsinized cells (total 0.5 × 10[5] cells) were then replated and treated with the same siRNA.

vector using the following pairs of primers: 5′-ATGCATCACCATCAC-CATCACATGGCGGCCATCCGCAAGAA-3′ (His-RhoB sense primer) and 5′-TCATAGCACCTTGCAGCAGT-3′ (RhoB antisense primer). The PCR products were introduced into the blunt ends of the pEU vector (Cell Free Science). The CNKSR1 gene was amplified from a vector containing the human CNKSR1 cDNA (FHC09975; Promega) using the following pairs of primers: 5′-ATGGAACCGGTAGAGACCTG-3′ (CNKSR1 sense primer) and 5′-GAGGTCAGGAGGTCGGAGGC-3′ (CNKSR1 antisense primer). The PCR products were introduced into the blunt ends of the pEU-C-bls vector (a king gift from Ms Yuki Tanaka, Ehime University). The PH domain of CNKSR1 gene was amplified from a vector containing the human CNKSR1 cDNA using the following pairs of primers: 5′-ATGCGGCCGGACTGTGACGG-3′ (PH sense primer) and 5′-TTAGTACTTGGAGATGCAGG-3′ (PH antisense primer). The PCR products were introduced into the blunt ends of the pEU-N-bls vector. The Myc-CNKSR1, CNKSR1 and Myc-PH domain of CNKSR1 gene were amplified from a vector containing the human CNKSR1 cDNA using the following pairs of primers: 5′-ATGGAACAAAAACTCATCTCAGAA-GAGGATCTGATGGAACCGGTAGAGACCTG-3′ (Myc-CNKSR1 sense primer) and 5′-TTAGAGGTCAGGAGGTCGGA-3′ (CNKSR1 antisense primer), 5′-ATGGAACCGGTAGAGACCTG-3′ (CNKSR1 sense primer), and 5′-TTA-GAGGTCAGGAGGTCGGA-3′ (CNKSR1 antisense primer), 5′-ATGGAA-CAAAAACTCATCTCAGAAGAGGATCTGATGCGGCCGGACTGTGACGG-3′ (Myc-PH sense primer) and 5′-TTAGTACTTGGAGATGCAGG-3′ (PH antisense primer). The PCR products were introduced into the blunt ends of the CSII-CMV-MCS-IRES2-Bsd vector (a king gift from Dr Hiroyuki Miyoshi, RIKEN). The deletion mutants and a W493A mutant of CNKSR1 were generated with 5′-AGCTCCAGGCTACAGACAGA-3′ (ΔSAM sense primer) and 5′-GGTCTCTACC-GGTTCCAT-3′ (ΔSAM antisense primer), 5′-GAGAAGGAGGGCACAGTCCT-3′ (ΔCRIC sense primer) and 5′-CTCTGTCTGTAGCCTGGAGC-3′ (ΔCRIC antisense primer), 5′-ATACCGGAGACCCCCCCACA-3′ (ΔPDZ sense primer) and 5′-CTGTTCCAGCAGCTCCTTGG-3′ (ΔPDZ antisense primer), 5′-CAGTCTC-CAGGCCGGGCCCC-3′ (ΔPH sense primer) and 5′-GCCCAGCTCACGGCAT-GACA-3′ (ΔPH antisense primer), 5′-CCTGAGCTGACAGGAGAGAA-3′ (ΔCC sense primer) and 5′-AGGGTCTCGGTTGCGCCGCA-3′ (ΔCC antisense primer), 5′-AGCATGGCGGTGCGTCATCTCATTACCTGC-3′ (W493A sense primer) and 5′-ACGCACCGCCATGCTCAGATCTGTCAGGGT-3′ (W493A antisense primer). The K414R and K414Q mutations were introduced using the following pairs of primers: 5′-TTGCGAAGGGCACCGGGCGGCTTCATGGGC-3′ (K414R sense primer) and 5′-CGGTGCCCTTCGCAACAGGAGCCAGCCGTC-3′ (K414R antisense primer), 5′-GTTGCGACAGGCACCGGGCGGCTTCATGGG-3′

(K414Q sense primer) and 5′-GGTGCCTGTCGCAACAGGAGCCAGCCGTCA-3′ (K414Q antisense primer). The gene encoded a K414E/P416E/R423A/R425L/R426A/K478N (EEALAN) mutant of PH domain (Indarte et al, 2019) was purchased from Eurofins Genomics. The PTPRH-HA gene was amplified from a vector containing the human PTPRH cDNA provided by the Kazusa DNA research institute (Nagase et al, 2008) using the following pairs of primers: 5′-ATGGCTGGGGCTGGCGGGGG-3′ (PTPRH sense primer) and 5′-TTAAGCGTAATCTGGAACATCGTATGGGTACATGACCTCCAACTTGTGGGCCT-3′ (PTPRH-HA antisense primer). The PCR products were introduced into the blunt ends of the CSII-CMV-MCS-IRES2-Bsd vector (a king gift from Dr Hiroyuki Miyoshi, RIKEN). The C1020S mutant of PTPRH were generated with 5′-GTGCACTCCAGTGCTGGCGTGGGTCGC-3′ (C1020S sense primer), 5′-AGCACTGGAGTGCACAATGGGTGGGCC-3′ (C1020S antisense primer). The gene of cytosolic region of PTPRH (PTPRH-cyto) was amplified from a vector containing the human PTPRH cDNA using the following pairs of primers: 5′-ATGCTGAAGAGGAGGAATAA-3′ (PTPRH-cyto sense primer) and 5′-GACCTCCAACTTGTGGGCCT-3′ (PTPRH-cyto antisense primer). The PCR product was introduced into the blunt ends of the pEU-C-FLAG vector (a king gift from Ms Yuki Tanaka, Ehime University). The deletion mutants of PTPRH were generated with 5′-CTCTCCCTGGTGGGCCACAGCCAGT-3′ (del-1 sense primer) and 5′-CAG-CATATCTTGGTGATGTAGATAG-3′ (del-1 antisense primer), 5′-GCC-AGCTTCATGCCCGGTCTCTGGA-3′ (del-2 sense primer) and 5′-TTGCTGGTACTCGTCTGCAAAACCA-3′ (del-2 antisense primer), 5′-GTGAAGTGTGAGCATTACTGGCCTC-3′ (del-3 sense primer) and 5′-ATTGATGTAGTCAGAGCCTGGCTCC-3′ (del-3 antisense primer), 5′-CGCCAATTCCACTACCAGGCCTGGC-3′ (del-4 sense primer) and 5′-CCGGCCGGCCTCCATGCAGTTGGTC-3′ (del-4 antisense primer), 5′-ACAGGAACCCTCATTGCCCTGGACG-3′ (del-5 sense primer) and 5′-CACAGACAGTGTCTTCTGCTCCTCC-3′ (del-5 antisense primer), 5′-CTGCGGTTCCTCCAACAGTCAGCCC-3′ (del-6 sense primer) and 5′-GCGACCCACGCCAGCACTGCAGTGC-3′ (del-6 antisense primer), 5′-ATCACTAGTGATTACAAGGATGACG-3′ (del-7 sense primer) and 5′-GATGCACTGATGCAGGAATACGTAC-3′ (del-7 antisense primer). The dihydrofolate reductase (DHFR)-FLAG-pEU and bls-DHFR-pEU vectors are kind gifts from Dr Tatsuya Sawasaki (Ehime University). The EGFR-GFP vector is a kind gift from Dr Shinji Fukuda (Aichi Gakuin University). The GFP-PLC-PH (pEGFP-C1) vector is a king gift from Dr Gregory Fairn (Dalhousie University, Canada). The CSII-CMV-MCS-IRES2-Bsd vectors carrying FLAG-CUL3 (Maekawa et al, 2017) or KCTD10 (Kovačević et al, 2018) were used as reported previously.

---

Cell number was counted 72 h after replating. Data are mean ± SEM from three independent experiments. **P < 0.01, *P < 0.05. **(D)** Western blots of cell-free synthesized FLAG-tagged cytosolic region of PTPRH. **(E)** In vitro binding assay for determination of the biotinylated proteins/PTPRH-cyto-FLAG interaction using AlphaScreen technology. Proteins synthesized by wheat germ extracts (Figs 2E and 7D) were subjected to AlphaScreen as indicated. DHFR was used as a negative control. The red lines indicate the threshold of interactions (10 times the luminescence signal for PTPRH-cyto/DHFR). Data from three independent experiments are expressed as the means ± SEM. **(F)** SKBR-3 cells stably expressing PTPRH-HA (wild-type or C1020S) were transfected with the indicated siRNAs. Cell lysates were prepared at 72 h post-siRNA transfection and subjected to Western blots. pEGFR, phosphorylated EGFR (Tyr[1068]); pHER2, phosphorylated HER2 (Tyr[1221/1222]). The ratio of band intensities for pEGFR/EGFR or pHER2/HER2, normalized to control, was shown. **(G)** SKBR-3 cells stably expressing PTPRH (wild-type or C1020S) were transfected with the indicated siRNAs for 48 h. Trypsinized cells (total 0.5 × 10[5] cells) were then replated and treated with the same siRNA. Cell number was counted 72 h after replating. Data are mean ± SEM from three independent experiments. n.s., not significant, *P < 0.05. **(H)** Confocal images of SKBR-3 cells treated with control siRNA or KCTD10 siRNA #1 for 72 h. 48 h after transfection of PTPRH-HA vector, cells were fixed, permeabilized, and stained for HA and EGFR. Bars; 10 μm. **(I)** SKBR-3 cells were treated with control siRNA or KCTD10 siRNA #1 for 72 h. 48 h after transfection of the PTPRH-HA and CNKSR1 vector, cell lysates were prepared and subjected to IP with HA antibody. Cell lysates and immunoprecipitants were analyzed by Western blot. **(J)** In vitro binding assay for determination of the CNKSR1-biotin//PTPRH-cyto-FLAG interaction using AlphaScreen technology. The protein mixture was incubated with purified His-RhoB (Q63L) that mimics RhoB-GTP (500 ng) for 1 h. See also Fig S1. DHFR was used as a negative control. Data from four independent experiments are expressed as the means ± SEM. n.s, not significant, **P < 0.01.
Source data are available for this figure.

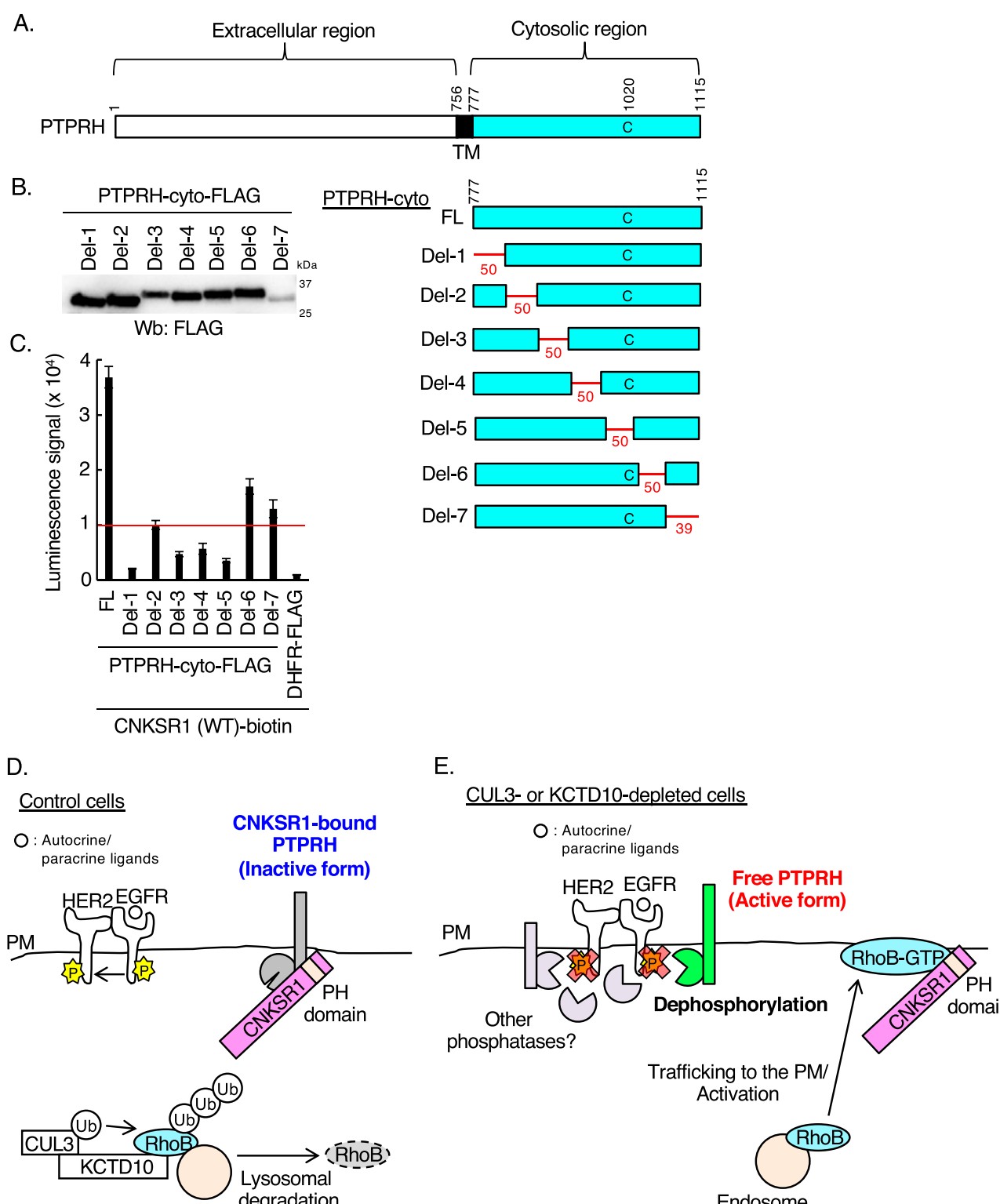

**Figure 8. In vitro binding assay for determination of interacting regions of PTPRH with connector enhancer of kinase suppressor of Ras1 (CNKSR1) using AlphaScreen technology.**

**(A)** The domain structure of human PTPRH. The numbers in the domain structure represent the order of amino acids from the N terminus. TM, transmembrane. The red lines indicate deleted regions in each deletion mutant. Fifty amino acids were deleted in del-1, 2, 3, 4, 5, and 6, and 39 amino acids were deleted in del-7 as shown in red. FL: full length of PTPRH-cyto-FLAG. The Cys[1020] residue is a phosphatase-active site. **(B)** Western blots of cell-free synthesized deletion mutants of FLAG-tagged cytosolic region of PTPRH. **(C)** In vitro binding assay for determination of the biotinylated CNKSR1/various mutants of PTPRH-cyto-FLAG interaction using AlphaScreen technology.

## Preparation of protein arrays

Human protein array used in this study consists of 4,212 human proteins fused with a FLAG-GST tandem tag at the N-terminus. The gene resources for the protein array were provided by the Kazusa DNA research institute (Nagase et al, 2008). Each cDNA clone was subcloned into pEU-E01-FLAG-GST-K1-02 vector (Yamanaka et al, 2021). The phosphatase focused array comprises 154 full-length phosphatases and 17 phosphatase catalytic domains. All proteins were fused with a FLAG-GST tandem tag at the N-terminus. From human protein array described above, 152 phosphatase clones were selected. The genes for the other phosphatases and phosphatase domains were inserted into the pEU-FLAG-GST-GW vector using seamless cloning using Gibson Assembly Master Mix (NEB). cDNA clone of PTPN9 and PTPN14 were from mammalian gene collection (Strausberg et al, 2002). DNA fragments coding phosphatase catalytic domains were amplified by PCR using cDNA clones (cDNA resources from Kazusa DNA research institute), mammalian gene collection [Strausberg et al, 2002], and gene synthesis [Eurofins]. Using diluted glycerol stocks containing each pEU expression plasmid as template, transcription template DNA fragments were amplified by PCR with PrimeStar Max PCR polymerase (Takara Bio), SPu-2 primer (5′-CAGTAAGCCAGATGCTACAC-3′) and AODA2306 primer (5′-AGCGTCAGACCCCGTAGAAA-3′). Transcription and translation reactions were conducted using WEPRO 7240 expression kit (CellFree Sciences). Transcription reaction mixture was prepared by mixing 2.5 $\mu$l of transcription buffer LM, 1.25 $\mu$l of NTP mixture (25 mM each), 0.25 $\mu$l RNase Inhibitor, 0.5 $\mu$l SP6 polymerase, and 2.5 $\mu$l PCR product in 96-well plate. The transcription reaction was incubated at 37°C for 18 h. Translation reaction mixture containing 2.5 $\mu$l of mRNA, 1.67 $\mu$l of WEPRO 7240 wheat germ extract, 0.14 $\mu$l of creatine kinase (20 mg/ml), and 0.11 $\mu$l RNase Inhibitor was prepared and overlaid with 44 $\mu$l of SUB-AMIX SGC in V-bottom 384 well plate. The translation reaction was incubated at 26°C for 18 h. The translated protein arrays were diluted twice with AlphaScreen buffer (100 mM Tris–HCl, pH 8.0, 0.01% Tween 20, 1 mg/ml BSA), divided into 384 plates, frozen in liquid nitrogen, and stored at –80°C. Expression of each N-FLAG-GST tagged protein was confirmed by Western blotting using anti-DYKDDDDK tag antibody (FUJIFILM Wako Pure Chemical).

## High throughput protein–protein interaction screen using AlphaScreen

All AlphaScreen reactions were conducted in AlphaPlate 384 titer plates (PerkinElmer). 20 ml of solution containing 0.5 $\mu$l of a biotin-tagged bait protein and 19.5 $\mu$l of the AlphaScreen buffer was dispensed into reaction plate by using automated multichannel pipet Viaflo and Viaflo Assist system (Integra). Next, 1 $\mu$l of each

FLAG-GST tagged protein array protein was transferred from 384-well stock plate to the reaction plate using Janus automated dispensing workstation (PerkinElmer) and Nanohead, a 384-well micro syringe head (PerkinElmer). Then 9 $\mu$l of detection mixture containing 0.02 $\mu$l of anti-DYKDDDK tag monoclonal antibody (Wako), 0.06 $\mu$l of streptavidin-conjugated AlphaScreen donor beads and 0.06 $\mu$l of protein A–conjugated AlphaScreen acceptor beads (PerkinElmer) in the AlphaScreen buffer were added to the reaction plates using FlexDrop dropper. After incubation at 25°C for 1 h in a dark incubator, AlphaScreen signal was detected by Envision multilabel reader (PerkinElmer).

## Production of recombinant proteins by wheat germ extracts and protein purification

Transcription and translation for the production of biotinylated or FLAG-tagged recombinant proteins were performed using the WEPRO 7240 Expression Kit (Cell Free Science) as reported previously (Tanigawa et al, 2019). For purification of RhoB-GTP, His-RhoB (Q63L) recombinant protein was produced using the WEPRO7240H Expression Kit (Cell Free Science). The crude proteins were incubated with Ni Sepharose High Performance resin (Cytiva), and His-RhoB-GTP was eluted with elution buffer (20 mM phosphate, 300 mM NaCl, and 500 mM imidazole, pH 7.5). The eluted proteins were dialyzed in suitable buffers.

## AlphaScreen

The in vitro binding assays were performed as previously described using an AlphaScreen IgG (Protein A) detection kit (Perkin Elmer) (Tezuka-Kagajo et al, 2020) with slight modifications. Briefly, 20 $\mu$l of detection mixture containing FLAG-tagged recombinant proteins, biotinylated recombinant proteins, 100 mM Tris–HCl (pH 7.5), 0.01% Tween 20, 1 $\mu$g/ml anti-FLAG antibody (Wako), 1 mg/ml BSA, 0.06 $\mu$l of streptavidin-coated donor beads and 0.06 $\mu$l of anti-IgG acceptor beads were added to each well of a 384-well Optiplate followed by incubation at room temperature for 1 h. Luminescence was detected by the AlphaScreen detection program using EnSpire (Perkin Elmer). The di-C8:0-PI(4,5)P$_2$ (Sigma-Aldrich) or PHT-7.3 (a CNKSR1 inhibitor; MedChemExpress) was added to the protein mixtures followed by incubation for 1 h.

## Cell culture

SKBR-3 cells, A431 cells, HepG2 cells, MIA-PaCa-2 cells, and HEK293T cells were maintained at 37°C with 5% CO2 in DMEM (Wako) supplemented with 10% FBS, 20 U/ml penicillin, and 100 $\mu$g/ml streptomycin. MCF-7 cells were maintained at 37°C with 5% CO2 in EMEM (Wako) supplemented with 10% FBS, 20 U/ml penicillin, and 100 $\mu$g/

---

Proteins synthesized by wheat germ extracts (Figs 2E, 7D, and 8B) were subjected to AlphaScreen as indicated. DHFR was used as a negative control. The red line indicates the threshold of interactions (10 times the luminescence signal for CNKSR1/DHFR). Data from three independent experiments are expressed as the means ± SEM. **(D, E)** Scheme of the present study. **(D)** In control SKBR-3 cells, CUL3/KCTD10 E3 complex constitutively ubiquitinates RhoB leading to its lysosomal degradation. In this condition, CNKSR1 forms a complex with PTPRH at the plasma membrane to inactivate its phosphatase activity for epidermal growth factor receptor. **(E)** Upon CUL3 or KCTD10 depletion, accumulated RhoB-GTP interacts with CNKSR1 at the plasma membrane resulting in the release of activated PTPRH from CNKSR1, which dephosphorylates epidermal growth factor receptor. PM, plasma membrane.
Source data are available for this figure.

ml streptomycin. MDA-MB-231 cells and MDA-MB-453 cells were maintained at 37°C without CO2 in Leiboviz's L-15 medium (Wako) supplemented with 10% FBS, 20 U/ml penicillin, and 100 μg/ml streptomycin. To inhibit neddylation, cells were treated with 1 μM MLN4924 (Boston Biochem) at 37°C for 24 h. Cells were treated with 1 μg/ml Rho inhibitor I (C3 transferase; Cytoskelton) at 37°C for 24 h, or with 50 μM PHT-7.3 (a CNKSR1 inhibitor; MedChemExpress) at 37°C for 24 h.

### Transfection of plasmid and siRNA

For transfection of plasmids into SKBR-3 cells and HEK293T cells, Lipofectamine 3000 (Invitrogen), and GeneJuice (Millipore) was used according to the manufacturer's instructions, respectively. At 48 h post-transfection, the cells were processed for subsequent experiments. Transfections of siRNAs (10–25 nM) into breast cancer cell lines were performed using RNAimax (Invitrogen) according to the manufacturer's instructions. Subsequent experiments were performed at 72 h post-transfection.

### Lentiviral expression

Transient or stable expression of Myc-CNKSR1 (WT, ΔPH, and W493A), Myc-PH, PTPRH-HA (WT, C1020S) in SKBR-3 cells was induced through lentiviral infection as previously described (Watanabe et al, 2020). The CSII-CMV-MCS-IRES2-Bsd, pCAG-HIVgp, and pCMV-VSVG-RSV-Rev vectors were kind gifts from Dr. Hiroyuki Miyoshi (RIKEN).

### siRNAs

The following validated siRNA duplex oligomers were purchased and used for knockdown experiments: GAGUGUAUGAGUUCCUAUU (siCUL3 #1; Sigma-Aldrich), GGAUCGCAAAGUAUACACAUAUGUA (siCUL3 #2; Invitrogen), GAAUGAGCGUCUAAAUCGU (siKCTD10 #1, targeting 3'-UTR of human KCTD10 mRNA; Sigma-Aldrich), GUAACAACAAAUACUCAUA (siKCTD10 #2; Sigma-Aldrich), GCAUCCAAGCCUACGACUA (siRhoB #1; Sigma-Aldrich), CUAAGAUGGUGUUAUUUAA (siRhoB #2, targeting 3'-UTR of human RhoB mRNA; Sigma-Aldrich), AGGCUACAGACA-GAGAACCUGCAAA (siCNKSR1 #1; Invitrogen), CAGAUCUGAGCAU-GUGGGUGCGUCA (siCNKSR1 #2; Invitrogen), GAGACAAACUAAUUGAGAA (siPPP1R8 #1; Thermo Fisher Scientific), GAGAAGAAGUAUUACUUAU (siPPP1R8 #2; Thermo Fisher Scientific), GCAUCAAAACUUAAUCAGA (siILKAP #1; Thermo Fisher Scientific), CAUGCAGCCUUAAGCCUCA (siILKAP #2; Thermo Fisher Scientific), GAACCGCUACAAGAACAUU (siPTPN6 #1; Thermo Fisher Scientific), ACCUCUCCCUGACCCUGUA (siPTPN6 #2; Thermo Fisher Scientific), CCAUUGUCACGGGCUUGUA (siENPP3 #1; Thermo Fisher Scientific), CAGACUUAUUGUAACAAGA (siENPP3 #2; Thermo Fisher Scientific), GGAAGAACCGCUACAAAGA (siPTPN18 #1; Thermo Fisher Scientific), AGAGCAGAUUCAAGAAAGA (siPTPN18 #2; Thermo Fisher Scientific), GGCUUGUCGCGAAUCUUCA (siG6PC3 #1; Thermo Fisher Scientific), CCCUAAAUCUGCUUCCGCA (siG6PC3 #2; Thermo Fisher Scientific), GGAUAAGAGGCAUGAGGAA (siPPP1R14C #1; Thermo Fisher Scientific), AGAGAGAGCUUCAAAAUUA (siPPP1R14C #2; Thermo Fisher Scientific), AGAUGAACGAUGACCACAA (siPPFIA3 #1; Thermo Fisher Scientific), GGCUCAACUAUGACCGGAA (siPPFIA3 #2; Thermo Fisher Scientific), AGAAAGGCCUGUUCAACUA

(siDUSP10 #1; Thermo Fisher Scientific), AGUUCGAGGAAGACCUAAA (siDUSP10 #2; Thermo Fisher Scientific), GGUACGAGAUGACAAGACA (siPPFIA1 #1; Thermo Fisher Scientific), GGUCGUUUUAGAUCAAUGA (siPPFIA1 #2; Thermo Fisher Scientific), AGUCCAACAUCUCGCCCAA (siDUSP7 #1; Thermo Fisher Scientific), ACAAGUUUCAAACAGAGUA (siDUSP7 #2; Thermo Fisher Scientific), GGGUCCAACGUAUCUCUUA (siPTPRR #1; Thermo Fisher Scientific), CAUUCGAAACCUUGUCUUA (siPTPRR #2; Thermo Fisher Scientific), GGACUACACCUACUGGGUA (siPTPRH #1; Thermo Fisher Scientific), GACUGAGGCUCAGUACGUA (siPTPRH #2; Thermo Fisher Scientific), GCCUAAUUCCUAUUACCCA (siPTPRM #1; Thermo Fisher Scientific), GGAAUAUCAUUGCAUACGA (siPTPRM #2; Thermo Fisher Scientific), GCACGUAUGACAAAGCGAU (siPTPRJ #1; Thermo Fisher Scientific), CAAGGACCUUUACCGAACA (siPTPRJ #2; Thermo Fisher Scientific). Control siRNAs were purchased from Sigma-Aldrich (SIC-001).

### Cell proliferation assay

A total of $1 \times 10^5$ SKBR-3 cells were seeded into a six-well plate in triplicate. Cells were treated with siRNA on the next day. After 48 h, $0.5 \times 10^5$ cells were replated into a 12-well plate in triplicate. Cells were then treated with the same siRNA again on the next day. The cells were counted at 72 h post-siRNA transfection.

### Western blotting and immunoprecipitation

Western blotting and immunoprecipitation were performed as previously described (Maekawa et al, 2017). The biotinylated proteins were detected using the VECTASTAIN ABC kit (Funakoshi).

### Pull-down assay

To detect GTP-Rho forms, SKBR-3 cells plated into a six-well plate were transfected with the indicated siRNAs. After 72 h, cells were lysed and subjected to pull-down assay to detect active GTP form of Rho using the RhoA Pull-down Activation Assay Biochem Kit (BK036; Cytoskeleton). To detect the interaction between RhoB-GTP and CNKSR1, SKBR-3 cells expressing Myc-CNKSR1 (WT, ΔPH or W493A) were lysed in IP buffer (25 mM Tris–HCl, pH 7.4, 150 mM NaCl, 1% NP-40, 1 mM EDTA, and 5% glycerol) containing cOmplete protease inhibitors (Roche). After incubation of cell lysates on ice for 10 min, the cell lysates were centrifuged at 10,000$g$ for 10 min at 4°C. The resulting supernatants were incubated with purified His-tagged RhoB-GTP recombinant proteins for 2 h at 4°C followed by incubation with prewashed TALON resin beads (Clontech) for 1 h at 4°C. The beads were washed three times with IP buffer.

### RT–PCR

Total RNAs were extracted from SKBR-3 cells using ISOGEN II (Nippon Gene) according to the manufacturer's protocol. The total RNA (1 μg) was used for cDNA synthesis using High Capacity RNA-to-cDNA Master Mix (Applied Biosystems). Real-time PCR was carried out (FastStart Universal SYBR Green Master ROX; Roche) on the ABI 7500 Real-Time PCR system (Applied Biosystems) using the following pairs of primers:

5'-GTCTGCCATGCCTTGTGCTC-3' (*EGFR* sense primer)
5'-CTTGTCCACGCATTCCCTGC-3' (*EGFR* antisense primer), 5'-AGCCTTGCCCCATCAACTG-3' (*HER2* sense primer), 5'-AATGCCAAC-CACCGCAGA-3' (*HER2* antisense primer), 5'-CATGAGAAGTATGACAA-CAGCCT-3' (*GAPDH* sense primer), 5'-AGTCCTTCCACGATACCAAAGT-3' (*GAPDH* antisense primer).

### Immunofluorescence staining

Cells were fixed with 4% PFA in PBS for 30 min at room temperature and permeabilized with 0.1% Triton X-100 in PBS for 15 min at room temperature. After blocking with 3% BSA in PBS for 30 min at room temperature, the cells were incubated with primary antibodies and then with secondary antibodies conjugated to fluorophores.

### Confocal microscopy and fluorescent recovery after photobleaching (FRAP)

Confocal microscopy was performed using the A1R laser confocal microscope (Nikon) with 60 × 1.27 Plan-Apochromat water immersion lens. Images were analyzed with FIJI (NIH). FRAP were performed on an A1R laser confocal microscope (Nikon), and FRAP analysis of EGFR-GFP was performed as described previously (Kay et al, 2012).

### Immunohistochemistry and hematoxylin and eosin staining

Immunohistochemical staining of HER2-positive breast cancer tissues was carried out as described previously (Tanigawa et al, 2019) with slight modifications. Briefly, the formaldehyde-fixed paraffin-embedded specimens were subjected to sectioning. Depar-affinized sections were incubated with 3% hydrogen peroxidase solution for 10 min at room temperature to block endogenous peroxidase activity followed by incubation with EDTA buffer (pH 9.0) for 10 min at 120°C to retrieve the antigen. After blocking with 3% BSA in PBS for 30 min at room temperature, the sections were incubated with primary antibodies at 4°C overnight and then with the secondary antibody (Histofine Simple Stain MAX-PO; Nichirei Biosciences Inc.) at room temperature for 45 min followed by the addition of DAB. The sections were also stained with hematoxylin for 1 min. Hematoxylin and eosin staining was carried out as described previously (Kiyoi, 2018). The stained specimens were observed using the all-in-one fluorescence microscope BIOREVO BZ-9000 (KEYENCE). The images were analyzed with the BZ-II analyzer (KEYENCE) and Fiji (NIH). We defined breast cancer cells as CNKSR1-positive when the signals for the CNKSR1-DAB staining were positive above the threshold 10 after subtraction. Disease-free survival and follow-up period according to CNKSR1-positive breast cancer cells were estimated using the Kaplan-Meier method. All statistical analyses were performed with the EZR software (Saitama Medical Center), version 3.6.1 (Kanda, 2013).

### Human breast cancer samples

All human HER2-positive breast cancer samples were collected from the archives of the Division of Diagnostic Pathology, Ehime University Hospital. Clinical and pathological information was obtained from the pathology records at Ehime University Hospital. This study was approved by the Institutional Review Board of Ehime University Hospital (approved IRB protocol number: 1701015). In-formed consent was obtained from all subjects. All experiments were performed by following the approved study plan and relevant guidelines.

### Clinical database analysis

Clinical datasets (The molecular taxonomy of breast cancer international consortium: METABRIC, EGA00000000083) deposited in the European Genome-phenome Archive (EGA) database were subjected to survival analysis. Survival analysis was performed by Kaplan–Meier method with overall survival status in R (ver. 3.6.3) as reported previously (Murakami et al, 2019).

### Expression analysis of HER2-positive breast cancer cell lines

RNA-seq datasets of HER2-positive breast cancer cell lines were downloaded from the Cancer Cell Line Encyclopedia (GSE36133) (Barretina et al, 2012). Expression data for PTPRH, CNKSR1, and EGFR extracted from the RNA-seq data were subjected to the correlational analysis and the heat map analysis with "ggplot2" and "pheatmap" packages in R (ver. 3.6.3).

### Statistical analyses

Statistical comparisons were made using the one-way ANOVA followed by Turkey's post hoc test. The log-rank test in survival analysis was performed by the "survminer" package in R (ver. 3.6.3.). Cox test in correlational analysis was performed in R.

# Supplementary Information

# Acknowledgements

We thank Ms. Ai Yanase, Ms Yoko Ikeda, Ms Mari Makimoto (Ehime University) for providing their technical assistance, Drs Igor Kovacevic (Martin Luther University Halle-Wittenberg), and Peter L Hordijk (Amsterdam University) for providing their useful information, Dr Masaaki Sawa (Carna Biosciences, Inc) for providing reagents. The establishment and screen of the protein arrays used in this study were partially supported by the Ministry of Education, Culture, Sports, Science and Technology, Japan (MEXT). Some of the cDNA resources were provided by a collaboration with the Kazusa DNA Research Institute. This work was supported by JSPS KAKENHI grant Number 19K18030 to K Nishiyama, Japan Foundation for Applied Enzymology, Research Grant of the Princess Takamatsu Cancer Research Fund, The Research Foundation for Pharmaceutical Sciences to M Maekawa, JSPS KAKENHI Grant Number 19K18029 to A Murakami, JSPS KAKENHI grant Number 21H02763 to S Higashiyama, Takeda Science Foundation to Proteo-Science Center.

### Author Contributions

K Nishiyama: data curation, formal analysis, funding acquisition, validation, investigation, and writing—original draft, review, and editing.

M Maekawa: conceptualization, data curation, formal analysis, supervision, funding acquisition, validation, investigation, project administration, writing—original draft, review, and editing, and co-first author.

T Nakagita: resources, data curation, validation, investigation, and methodology.

J Nakayama: resources, software, formal analysis, and investigation.

T Kiyoi: data curation, formal analysis, and validation.

M Chosei: formal analysis, validation, and investigation.

A Murakami: formal analysis, funding acquisition, validation, and investigation.

Y Kamei: data curation, formal analysis, supervision, and validation.

H Takeda: conceptualization, resources, data curation, formal analysis, methodology, and writing—original draft, review, and editing.

Y Takada: data curation, formal analysis, supervision, validation, and writing—original draft, review, and editing.

S Higashiyama: conceptualization, supervision, funding acquisition, validation, investigation, project administration, and writing—original draft, review, and editing.

## Conflict of Interest Statement

The authors declare that they have no conflict of interest.

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
