## [Reviewer comments · Life Science Alliance]

Life Science Alliance

CNKSR1 serves as a scaffold to activate an EGFR phosphatase via exclusive interaction with RhoB-GTP

Kanako Nishiyama, Masashi Maekawa, Tomoya Nakagita, Jun Nakayama, Takeshi Kiyoi, Mami Chosei, Akari Murakami, Yoshiaki Kamei, Hiroyuki Takeda, Yasutsugu Takada, and Shigeki Higashiyama

DOI: <https://doi.org/10.26508/lsa.202101095>

Corresponding author(s): Masashi Maekawa, Division of Cell Growth and Tumor Regulation, Proteo-Science Center (PROS), Ehime University and Shigeki Higashiyama, Ehime University

Review Timeline:

Submission Date:	2021-04-15
Editorial Decision:	2021-06-01
Revision Received:	2021-06-15
Editorial Decision:	2021-06-17
Revision Received:	2021-06-17
Accepted:	2021-06-21

Transaction Report:

June 1, 2021

Re: Life Science Alliance manuscript #LSA-2021-01095-T

Dr. Masashi Maekawa
Division of Cell Growth and Tumor Regulation, Proteo-Science Center (PROS), Ehime University
Toon
Ehime, Ehime 791-0295
Japan

Dear Dr. Maekawa,

Thank you for submitting your manuscript entitled "CNKSR1 serves as a scaffold to activate an EGFR phosphatase via exclusive interaction with RhoB-GTP" to Life Science Alliance. The manuscript was assessed by expert reviewers, whose comments are appended to this letter.

As you will note from the reviewers' comments below, the reviewers find your data interesting, but have also raised some questions that should be addressed prior to further consideration of the manuscript at LSA. We would, thus, like to invite you to submit a revised version of the manuscript that addresses all of the reviewers' points.

Thank you for this interesting contribution to Life Science Alliance. We are looking forward to receiving your revised manuscript.

Sincerely,

Shachi Bhatt, Ph.D.

Executive Editor
Life Science Alliance
<http://www.lsjournal.org>
Tweet @SciBhatt @LSAJournal

- A letter addressing the reviewers' comments point by point.
- An editable version of the final text (.DOC or .DOCX) is needed for copyediting (no PDFs).
- High-resolution figure, supplementary figure and video files uploaded as individual files: See our detailed guidelines for preparing your production-ready images, <https://www.life-science-alliance.org/authors>
- Summary blurb (enter in submission system): A short text summarizing in a single sentence the study (max. 200 characters including spaces). This text is used in conjunction with the titles of papers, hence should be informative and complementary to the title and running title. It should describe the context and significance of the findings for a general readership; it should be written in the present tense and refer to the work in the third person. Author names should not be mentioned.

B. MANUSCRIPT ORGANIZATION AND FORMATTING:

Reviewer #1 (Comments to the Authors (Required)):

The authors demonstrated that degradation by the ubiquitin ligase complex CUL3/KCTD10 of RhoB is important for EGFR and HER2 phosphorylation in HER2-positive breast cancer cells. They also showed that both RhoB-GTP and protein tyrosine phosphatase PTPRH interacted with CNKSR1. Degradation of RhoB makes CNKSR1 to interact with PTPRH at the plasma membrane, resulting in inactivation of EGFR phosphatase activity. By contrast, depletion of CUL3 or KCTD10 resulted in the accumulation of RhoB-GTP at the plasma membrane and its interaction with CNKSR1, which released activated PTPRH from CNKSR1 and promoted dephosphorylation by PTPRH of EGFRs. Thus, authors suggested that CNKSR1 works a molecular switch that

inactivates PTPRH in steady state but upon loss of CUL3/KCTD10 interacts with RhoB and releases active PTPRH.

The data was well presented and discussed extensively. As the PTPRH is exclusively expressed in the intestinal epithelial cells, the author could show whether the same mechanism could work in normal intestinal epithelial cells or only certain types of cells that overexpress PTPRH.

Reviewer #2 (Comments to the Authors (Required)):

In this study Nishiyama et al. identified the scaffold protein CNKSR1 as regulator of EGFR activity and propose a molecular mechanism underlying this process. Interaction with CNKSR1 prevents the phosphatase PTPRH to dephosphorylate its substrate EGFR correlating with sustained EGFR activity. The GTPase RHOB competes with PTPRH for binding to the PH domain of CNKSR1 leading to the release of PTPRH and subsequent dephosphorylation and deactivation of EGFR. The data of this comprehensive study are highly interesting and connect the oncoprotein CNKSR1 with the tumor suppressor PTPRH and the GTPase RHOB at least in SKBR-3 cells. However, to support the proposed molecular mechanism some questions have to be answered.

Major points:

1. Fig.1: The authors clearly demonstrate that knockdown of CUL3 or KCTD10 increase the level of RHOB in SKBR-3 cells. Is anything known about the mechanism regulating the expression level of CUL3 or KCTD10 under physiological conditions?
2. Fig. 2H: Have the authors any explanation for the strong interaction between Biotin-PH(EELAN) and RHOB-Q63L? Could binding of wild-type CNKSR1 to phosphoinositides via the PH domain interfere with binding to RHOB in cells?
3. Page 7, 1st paragraph: "RHOB-GTP directly interacts with CNKSR1 through recognition of the Trp493 ...". Which data show a direct interaction with Trp493? The mutant W493A may induce a conformational change of the PH domain.
4. Fig. 1B/Fig. 5F: Knockdown of CUL3 or KCTD10 seem to increase expression of EGFR and decrease expression of HER2. Fig. 3G: Knockdown of CNKSR1 seems to decrease the expression of HER2. Can the authors comment these effects?
5. Knockdown of CNKSR1 reduces pEGFR and pHER2. For the effect on pEGFR, the authors identified PTPRH as the respective phosphatase (Fig. 6, Fig. S4). Do the authors test the CNKSR1-interacting phosphatases on their effect on pHER2- similar as they did for pEGFR in Fig. S4?
6. Abstract, last sentence "unique model of phosphatase activation by protein-protein interaction". Is PTPRH activated by the release from CNKSR1 or is PTPRH a constitutively activated PTPRH and the substrate accessibility is restricted by binding to CNKSR1? Can PTPRH be regulated by posttranslational modifications?

Minor points:

1. Page 10: PTPRH, full name with abbreviation should be mentioned when used for the first time - in this case in the introduction.
2. Page 8: Change "HER2-potitive- type" to "HER2-positive-type"
3. Page 10, 2nd line: "Stimulation of EGFR .." should be replaced by "Stimulation of SKBR-3 cells ...".
4. Fig. 2E, legend: "FALG-tagged" should be corrected to "FLAG-tagged".

We are submitting our revised manuscript, “CNKSR1 serves as a scaffold to activate an EGFR phosphatase via exclusive interaction with RhoB-GTP” (LSA-2021-01095-T) for publication in the *Life Science Alliance*.

Thank you for very much your letter on June 1, 2021. We greatly appreciate the comments from the reviewers. Our responses to the comments are shown below. For reference, the reviewers' comments are shown in *italics*. In the main text, the revised points are shown in red.

As described in more detail below, we have experimentally addressed all the reviewers' concerns. With this extensive revision, we hope that the reviewers will concur with us that we have addressed all of the raised concerns in a satisfactory manner and, consequently, substantially strengthened our paper.

Thank you again for considering our work for publication.

Response to previous review

Reviewer 1:

The data was well presented and discussed extensively. As the PTPRH is exclusively expressed in the intestinal epithelial cells, the author could show whether the same mechanism could work in normal intestinal epithelial cells or only certain types of cells that overexpress PTPRH.

Response: Thank you very much for your valuable comments. As you mentioned, PTPRH is expressed in epithelial cells of the small intestine and stomach in mice (Murata et al 2015, Sadakata et al 2009). To examine if EGFR phosphorylation is regulated by RhoB degradation by the CUL3/KCTD10 E3 complex and CNKSR1 in intestinal epithelial cells, we cultured primary human intestinal epithelial cells (InEpC cells). However, it was technically very difficult to continue tissue cultures of the primary InEpC cells, and transfection of siRNA with RNAimax was highly toxic to the primary InEpC cells. The roles of CUL3/KCTD10 E3 complex and CNKSR1 in the PTPRH regulation in normal epithelial cells of the small intestine and stomach would be investigated using mice models. We shortly described the points in Discussion part (**page 15, line 420 – page 16, line 423**).

We next examined the effects of CUL3, KCTD10 or CNKSR1 knockdown on EGFR phosphorylation and RhoB expression in various cancer cell lines. As shown in Fig. S7, EGFR phosphorylation was detected in neither A431 cells (PTPRH^{high}), HepG2 cells (PTPRH^{low}) nor MIA-PaCa-2 cells (PTPRH^{low}) in the steady-state condition. These data suggest that EGFR is not activated by endogenously-produced ligands in those cancer cells. Knockdown of CUL3, but not KCTD10, increased the protein expression of RhoB in A431 cells (Fig. S7A), suggesting the contribution of other BTBPs for RhoB degradation in A431 cells. In HepG2 cells, CUL3 or KCTD10 knockdown caused the accumulation of RhoB protein as was seen in SKBR-3 cells (Fig. S7B). In contrast, RhoB protein was accumulated by knockdown of neither CUL3 nor KCTD10 in MIA-PaCa-2 cells (Fig. S7C). Taken together, it is likely that the regulation of EGFR phosphorylation through PTPRH activation and RhoB degradation by the CUL3/KCTD10 E3 complex is conserved in HER2/EGFR-double positive breast cancer cells. We described the data in Discussion part (**page 15, line 409 - line 420**).

Reviewer 2:

Major points:

1. Fig. 1: The authors clearly demonstrate that knockdown of *CUL3* or *KCTD10* increase the level of *RHOB* in SKBR-3 cells. Is anything known about the mechanism regulating the expression level of *CUL3* or *KCTD10* under physiological conditions?

Response: Thank you very much for your comments. During the physiological angiogenesis, hypoxia induced expression of a microRNA, miR-101, which binds to the 3' untranslated region of *CUL3* mRNA (Kim et al 2014). The miR-101 downregulates protein expression of *CUL3* in human umbilical vein endothelial cells, leading to the upregulation of VEGF and promotion of angiogenesis both *in vitro* and *in vivo* (Kim et al 2014). Other microRNAs, miR-455 and miR-601, which are induced by hydrogen peroxide, also target the 3' untranslated region of *CUL3* mRNA and downregulate *CUL3* expression in osteoblasts and retinal pigment epithelium cells, respectively (Chen et al 2019, Xu et al 2017). A previous literature suggests that miR-592 may downregulate *KCTD10* expression during the development of congenital heart diseases (Pang et al 2019). These data suggest that expression of *CUL3* and *KCTD10* would be reduced by microRNAs also in SKBR-3 cells. The identification of specific patho-physiological conditions in which expression of *CUL3* or *KCTD10* changes in HER2-positive breast cancer cells would be investigated in future. We described these points in Discussion part (**page 13, line 332 - line 342**).

2. Fig. 2H: Have the authors any explanation for the strong interaction between Biotin-PH(E_EALAN) and *RHOB-Q63L*? Could binding of wild-type *CNKSRI* to phosphoinositides via the PH domain interfere with binding to *RHOB* in cells?

Response: Thank you very much for pointing out. Regarding to the strong interaction between Biotin-PH (E_EALAN) and FLAG-RhoB (Q63L), it is likely that the three-dimensional structure of PH domain is largely changed by the introduction of six point-mutations (K414E/P416E/R423A/R425L/R426A/K478N) in the domain. RhoB-GTP may more easily access to its binding pocket in the mutant PH domain by conformational changes of the mutant. We shortly described the possibility in Results part (**page 7, line 153 - line 158**). From this standpoint, the structural analysis of RhoB-GTP/*CNKSRI* complex as well as PTPRH/*CNKSRI* complex remains to be investigated. Please also see the response to your comment 3 and 6 below. We discussed the points in Discussion part (**page 14, line 390 – page 15, line 392**).

To examine if phosphatidylinositol 4,5 bisphosphate (PI(4,5)P₂), an interacting phosphoinositide of *CNKSRI* (Indarte et al 2019), interferes the interaction between wild-type *CNKSRI* and RhoB-GTP in cells, we performed four additional experiments below.

(1) We confirmed that GFP-PLC-PH, a biosensor for PI(4,5)P₂, localized at the plasma membrane upon *CUL3* or *KCTD10* knockdown as was seen in control SKBR-3 cells (Fig. 4A). These data suggested that accumulation of RhoB at the plasma membrane by *CUL3* or *KCTD10* knockdown did not affect the plasmalemmal localization of PI(4,5)P₂.

(2) A previous literature showed that di-8:0-PI(4,5)P₂ directly binds to the PH domain of CNKSR1 *in vitro* (Indarte et al 2019). We showed that the addition of di-8:0-PI(4,5)P₂ into AlphaScreen of CNKSR1 (WT)-biotin/FLAG-RhoB (Q63L) did not affect the luminescence signals between CNKSR1 (WT)-biotin and FLAG-RhoB (Q63L) *in vitro* (Fig. 2I). These data suggest that PI(4,5)P₂ does not interfere the interaction between wild-type CNKSR1 and RhoB-GTP, at least, *in vitro*.

(3) We next sought to examine the effects of PI(4,5)P₂ on the interaction between CNKSR1 and RhoB-GTP in SKBR-3 cells. For this aim, we used a CNKSR1 inhibitor, PHT-7.3 (Indarte et al 2019). PHT-7.3 was identified as a CNKSR1-PH domain-binding compound, and the compound supposes to occupy the PI(4,5)P₂-binding pocket in the PH domain (Indarte et al 2019). Thus, upon treatment of PHT-7.3, CNKSR1 loses both binding capacity to PI(4,5)P₂ and its plasmalemmal localization. Actually, treatment of K-Ras mutated cancer cells with PHT-7.3 relocated CNKSR1 to the cytosol from the plasma membrane (Indarte et al 2019). Similarly, we found that CNKSR1 lost its localization at the membrane ruffles (specific structures of the plasma membrane indicated by arrowheads) in PHT-7.3-treated SKBR-3 cells (Fig. 4B). In contrast, upon KCTD10 knockdown, CNKSR1 localized at the plasma membrane even by the treatment with PHT-7.3, and colocalization with RhoB at the plasma membrane was observed in PHT-7.3-treated SKBR-3 cells (Fig. 4B). These data suggest that CNKSR1 can interact with RhoB at the plasma membrane in a PI(4,5)P₂-independent manner in KCTD10-knockdown cells.

(4) We detected enough luminescence signals for the direct interactions between CNKSR1 and RhoB-GTP in the presence of PHT-7.3 *in vitro* (Fig. 4C). These data suggest that a PI(4,5)P₂-binding site of the PH domain of CNKSR1 is not critical for the binding of RhoB-GTP to the PH domain.

Taken together, these results implicate that PI(4,5)P₂ does not interfere the interaction between wild-type CNKSR1 and RhoB-GTP in SKBR-3 cells. We described those data in an existing paragraph and a new paragraph in Result part (**page 7, line 158 - line 160, page 8, line 200 – page 9, line 219**).

3. Page 7, 1st paragraph: "RHOB-GTP directly interacts with CNKSR1 through recognition of the Trp493 ...". Which data show a direct interaction with Trp493? The mutant W493A may induce a conformational change of the PH domain.

Response: We really apologize our misreading of the data and inappropriate description. We fully agree with your opinion that RhoB-GTP do not directly interact with a Trp⁴⁹³ of the PH domain because introduction of the mutation could induce conformational changes of the domain. We thus revised the manuscript in Result part as follow; "the Trp⁴⁹³ residue in the PH domain of CNKSR1 is critical for the direct interaction between CNKSR1 and RhoB-GTP" (**page 7, line 167 - line 168**). Regarding to the conformational changes of the W493A mutant, the structural analysis of RhoB-GTP/CNKSR1 complex by X-ray crystallography or cryo-electron microscopy would provide critical information. We described the data and discussed in Discussion part (**page 14, line 390 – page 15, line 392**). Please also see the response to your comment 2 and 6.

4. Fig. 1B/Fig. 5F: Knockdown of CUL3 or KCTD10 seem to increase expression of EGFR and decrease

expression of HER2. Fig. 3G: Knockdown of CNKSR1 seems to decrease the expression of HER2. Can the authors comment these effects?

Response: Thank you very much for valuable comments. We examined mRNA expression of EGFR and HER2 in CUL3, KCTD10 or CNKSR1 depleted SKBR-3 cells. As shown in Fig. S5, knockdown of neither CUL3, KCTD10 nor CNKSR1 significantly affected the mRNA level of both EGFR and HER2. These data suggested that CUL3, KCTD10 and CNKSR1 may contribute to mRNA translation or/and protein turnover of EGFR and HER2. Previous literatures actually showed that CUL3 is essential for trafficking and degradation of EGFR (Gschweidl et al 2016, Huotari et al 2012). We described the data and discussed in Discussion part (**page 13, line 343 - line 350**).

5. Knockdown of CNKSR1 reduces pEGFR and pHER2. For the effect on pEGFR, the authors identified PTPRH as the respective phosphatase (Fig. 6, Fig. S4). Do the authors test the CNKSR1-interacting phosphatases on their effect on pHER2- similar as they did for pEGFR in Fig. S4?

Response: Thank you very much for valuable comments. We run the gels using the same cell lysates used in Fig. S4, and detected both pHER2 and HER2. As shown in revised Fig. S4, knockdown of any CNKSR1-interacting phosphatases did not obviously restore the decreased level of HER2 phosphorylation in KCTD10-depleted SKBR-3 cells. These data suggest the redundancy of CNKSR1-interacting phosphatases in the regulation of HER2 phosphorylation. We described the results and discuss possibility of the redundancy (**page 11, line 290 - line 293**).

6. Introduction, last sentence "unique model of phosphatase activation by protein-protein interaction". Is PTPRH activated by the release from CNKSR1 or is PTPRH a constitutively activated PTPRH and the substrate accessibility is restricted by binding to CNKSR1? Can PTPRH be regulated by posttranslational modifications?

Response: Thank you very much for valuable comments. We fully agree to the suggested two models of CNKSR1-mediated PTPRH activation. To address the molecular mechanisms of PTPRH activation, we sought to identify CNKSR1-interacting domains in the cytosolic region of PTPRH. We generated seven deletion mutants of PTPRH-cyto, which delete 50 or 39 sequential amino acids in cytosolic region of PTPRH (Fig. 8A, 8B). As shown in Fig. 8C, the *in vitro* binding assay by AlphaScreen indicated that luminescence signals between CNKSR1 and the cytosolic region of PTPRH reduced to below threshold in the del-1, del-3, del-4 and del-5 mutants. These data suggests that CNKSR1 may recognize a specific three-dimensional structure of the overall cytosolic region of PTPRH. Importantly, the deletion of amino acid sequences including a cysteine residue (Cys¹⁰²⁰), which is essential for phosphatase activities, resulted in the loss of capacity to interact with CNKSR1 (the del-5 mutant, Fig. 8C). In the PTPRH/CNKSR1 complex, the phosphatase-active site of PTPRH could be masked by CNKSR1, which inactivate PTPRH phosphatase activities because its active site cannot access to EGFR. Upon accumulation of RhoB-GTP, CNKSR1 forms a complex with RhoB-GTP, and then released PTPRH, which active site are open to access EGFR, may exert phosphatase activities. The structural analysis of RhoB-GTP/CNKSR1 complex as well as

PTPRH/CNKSR1 complex remain to be investigated in details by X-ray crystallography or cryo-electron microscopy. We described the data and discussed in Result and Discussion parts (**page 12, line 318 - line 324; page 14, line 383 - 393**). Please also see the response to your comment 2 and 3 above.

Regarding to the posttranslational modification of PTPRH, the extracellular motif of PTPRH is known to be glycosylated (Matozaki et al 1994). Another glycosylated protein tyrosine phosphatase receptor, PTPRC (alias CD45), reduces accessibility of large ligands (*e.g.* dead cells) to phagocytic receptors through the formation of “glycocalyx pickets” (Freeman et al 2016, Ostrowski et al 2016). PTPRC also limits mobility of the phagocytic receptors through the formation of cortical actin fences in cytosol (Freeman et al 2016, Ostrowski et al 2016). Thus, the glycosylation level of PTPRH may affect the accessibility of substrates (*e.g.* EGFR) to PTPRH. The glycosylated extracellular regions of PTPRH would function as “glycocalyx pickets” like PTPRC. The identification of responsible enzymes for PTPRH glycosylation as well as glycosylated amino acid residues are needed to elucidate the physiological meanings of glycosylation of PTPRH. We discussed the points in Discussion part (**page 14, line 397 - line 405**).

Minor points:

1. Page 10: PTPRH, full name with abbreviation should be mentioned when used for the first time - in this case in the introduction.

Response: We apologized our carelessness. We described the full name of PTPRH and CNKSR1 in the Introduction part (**page 4, line 87 – line 89**).

2. Page 8: Change "HER2-potitive- type" to "HER2-positive-type"

Response: We apologized the mistake. We revised as you mentioned (**page 9, line 230**).

3. Page 10, 2nd line: "Stimulation of EGFR ..." should be replaced by "Stimulation of SKBR-3 cells ...".

Response: We apologized the mistake. We revised as you mentioned (**page 11, line 275**).

4. Fig. 2E, legend: "FALG-tagged" should be corrected to "FLAG-tagged".

Response: We apologized the mistake. We revised as you mentioned (**page 37, line 1058**).

References

- Chen ZJ, Rong L, Huang D, Jiang Q. 2019. Targeting cullin 3 by mir-601 activates nrf2 signaling to protect retinal pigment epithelium cells from hydrogen peroxide. *Biochem Biophys Res Commun.* 515(4):679-687. doi:10.1016/j.bbrc.2019.05.171
- Freeman SA, Goyette J, Furuya W, Woods EC, Bertozzi CR, Bergmeier W, Hinz B, van der Merwe PA, Das R, Grinstein S. 2016. Integrins form an expanding diffusional barrier that coordinates phagocytosis. *Cell.* 164(1-2):128-140. doi:10.1016/j.cell.2015.11.048

Gschweidl M, Ulbricht A, Barnes CA, Enchev RI, Stoffel-Studer I, Meyer-Schaller N, Huotari J, Yamauchi Y, Greber UF, Helenius A, et al. 2016. A spop1/cullin-3 ubiquitin ligase complex regulates endocytic trafficking by targeting eps15 at endosomes. *Elife*. 5:e13841. doi:10.7554/eLife.13841

Huotari J, Meyer-Schaller N, Hubner M, Stauffer S, Katheder N, Horvath P, Mancini R, Helenius A, Peter M. 2012. Cullin-3 regulates late endosome maturation. *Proc Natl Acad Sci U S A*. 109(3):823-828. doi:10.1073/pnas.1118744109

Indarte M, Puentes R, Maruggi M, Ihle NT, Grandjean G, Scott M, Ahmed Z, Meuillet EJ, Zang S, Lemos R, Jr., et al. 2019. An inhibitor of the pleckstrin homology domain of cnk1 selectively blocks the growth of mutant kras cells and tumors. *Cancer research*. 79(12):3100-3111. doi:10.1158/0008-5472.Can-18-2372

Kim JH, Lee KS, Lee DK, Kim J, Kwak SN, Ha KS, Choe J, Won MH, Cho BR, Jeoung D, et al. 2014. Hypoxia-responsive microRNA-101 promotes angiogenesis via heme oxygenase-1/vascular endothelial growth factor axis by targeting cullin 3. *Antioxidants & redox signaling*. 21(18):2469-2482. doi:10.1089/ars.2014.5856

Matozaki T, Suzuki T, Uchida T, Inazawa J, Ariyama T, Matsuda K, Horita K, Noguchi H, Mizuno H, Sakamoto C, et al. 1994. Molecular cloning of a human transmembrane-type protein tyrosine phosphatase and its expression in gastrointestinal cancers. *J Biol Chem*. 269(3):2075-2081.

Murata Y, Kotani T, Supriatna Y, Kitamura Y, Imada S, Kawahara K, Nishio M, Daniwijaya EW, Sadakata H, Kusakari S, et al. 2015. Protein tyrosine phosphatase sap-1 protects against colitis through regulation of ceacam20 in the intestinal epithelium. *Proc Natl Acad Sci U S A*. 112(31):E4264-4271. doi:10.1073/pnas.1510167112

Ostrowski PP, Grinstein S, Freeman SA. 2016. Diffusion barriers, mechanical forces, and the biophysics of phagocytosis. *Dev Cell*. 38(2):135-146. doi:10.1016/j.devcel.2016.06.023

Pang XF, Lin X, Du JJ, Zeng DY. 2019. Downregulation of microRNA-592 protects mice from hypoplastic heart and congenital heart disease by inhibition of the notch signaling pathway through upregulating kctd10. *Journal of cellular physiology*. 234(5):6033-6041. doi:10.1002/jcp.27190

Sadakata H, Okazawa H, Sato T, Supriatna Y, Ohnishi H, Kusakari S, Murata Y, Ito T, Nishiyama U, Minegishi T, et al. 2009. Sap-1 is a microvillus-specific protein tyrosine phosphatase that modulates intestinal tumorigenesis. *Genes to cells : devoted to molecular & cellular mechanisms*. 14(3):295-308. doi:10.1111/j.1365-2443.2008.01270.x

Xu D, Zhu H, Wang C, Zhu X, Liu G, Chen C, Cui Z. 2017. MicroRNA-455 targets cullin 3 to activate nrf2 signaling and protect human osteoblasts from hydrogen peroxide. *Oncotarget*. 8(35):59225-59234. doi:10.18632/oncotarget.19486

June 17, 2021

RE: Life Science Alliance Manuscript #LSA-2021-01095-TR

Dr. Masashi Maekawa

Division of Cell Growth and Tumor Regulation, Proteo-Science Center (PROS), Ehime University
Japan

Dear Dr. Maekawa,

Thank you for submitting your revised manuscript entitled "CNKSR1 serves as a scaffold to activate an EGFR phosphatase via exclusive interaction with RhoB-GTP". We would be happy to publish your paper in Life Science Alliance pending final revisions necessary to meet our formatting guidelines.

- please add ORCID ID for secondary corresponding author-they should have received instructions on how to do so
- please make sure the author order in your manuscript and our system match
- please add callouts for Figures S3A, B, and S5A, B to your main manuscript text

A. FINAL FILES:

B. MANUSCRIPT ORGANIZATION AND FORMATTING:

Sincerely,

June 21, 2021

RE: Life Science Alliance Manuscript #LSA-2021-01095-TRR

Dr. Masashi Maekawa
Division of Cell Growth and Tumor Regulation, Proteo-Science Center (PROS), Ehime University
Toon
Ehime, Ehime 791-0295
Japan

Dear Dr. Maekawa,

Thank you for submitting your Research Article entitled "CNKSR1 serves as a scaffold to activate an EGFR phosphatase via exclusive interaction with RhoB-GTP". It is a pleasure to let you know that your manuscript is now accepted for publication in Life Science Alliance. Congratulations on this interesting work.

DISTRIBUTION OF MATERIALS:

Again, congratulations on a very nice paper. I hope you found the review process to be constructive and are pleased with how the manuscript was handled editorially. We look forward to future exciting submissions from your lab.

Sincerely,
